# Managing Phosphorus Loss from Agroecosystems of the Midwestern United States: A Review

**Gurbir Singh [1], Gurpreet Kaur [1,*], Karl Williard [2], Jon Schoonover [2] and Kelly A. Nelson [3]**

1   Delta Research and Extension Center, Mississippi State University, Stoneville, MS 38776, USA; gurbir.singh@msstate.edu
2   Department of Forestry, Southern Illinois University, Carbondale, IL 62901, USA; williard@siu.edu (K.W.); jschoon@siu.edu (J.S.)
3   Division of Plant Sciences, University of Missouri, Novelty, MO 63460, USA; NelsonKe@missouri.edu
*   Correspondence: gk340@msstate.edu; Tel.: +1-662-390-8507

**Abstract:** Best management practices (BMPs) are site-specific and their implementation, long-term management, and maintenance are important for successful reduction of phosphorus (P) loss into headwater streams. This paper reviews published research on managing P loss from agricultural cropping systems in the Midwestern United States and classified the available research based on BMPs and their efficacy in reducing P loss. This review paper also identifies the areas where additional research could provide insight for managing P losses. Our literature review shows that cover crops, reduced tillage, saturated buffers, and constructed wetlands are the most evaluated areas of current research. However, additional research is necessary on the site-specific area to measure the effectiveness of BMPs in managing P loss. The BMPs that serve as a sink of P need further evaluation in long-term field-scale trials. Studies evaluating adsorption and desorption mechanisms of P in surface and subsurface soils with materials or amendments that bind P in the soil are needed. The time required and pathways, where the flush of available P is lost or fixed in the soil matrix, need further investigation. Measured P loss from BMPs like bioreactors and saturated buffers supplemented with P adsorption materials or filters need to be simulated with models for their prediction and validation. Field evaluations of P index and critical source area concepts should be investigated for identifying problematic areas in the watersheds. Identification of overlapping areas of high P source and transport can help in strategic planning and layout, thereby resulting in reducing the cost of implementing BMPs at field and watershed scales.

**Keywords:** cover crops; tillage; terraces; water and sediment control basins; vegetative buffers

## 1. Introduction

Phosphorus is one of the most limiting nutrients for crop production after nitrogen. Cereal crops have only 16% P fertilizer use efficiency globally as estimated by Dhillon et al. [1]. Over application of P fertilizer to crops has resulted in an estimated loss of about 10.5 million metric tons of P each year from agricultural fields that are equal to half of the P mined every year [2]. Managing P losses from anthropogenic inputs into ecosystems is a global issue; therefore, several reviews have been published to address this issue [3–16]. Sharpley et al. [3] highlighted some important areas for potential P research including identification of soil P levels that have a high risk of P loss into water bodies, targeting critical source areas for managing P loss at watershed scales and balancing economically and environmentally sustainable use of P fertilizer.

In the last two to three decades, a fair amount of research has focused on developing a toolbox of best management practices (BMPs) for managing N and P. However, water quality improvements using

BMPs have shown mixed results. Reasons identified for the poor performance of implemented BMPs include legacy P inputs (P from a prior land application and nutrient management), climate fluctuations, ineffective conservation practices, mistakes in understanding pollution sources, poor experimental design, inadequate level or distribution of BMPs and inadequate P management policies [9,16]. In addition, poor experimental design in any research study might not be able to prove the efficiency of the BMP tested. Several BMPs have proven to reduce N and P losses from agricultural fields. These BMPs are classified into three categories including in-field, edge of the field and land-use changes. The BMPs for managing P losses have been explored and researched to a lesser extent compared to BMPs focused on N losses. The objectives of this article were to (1) review the published research on managing P loss from agricultural cropping systems in the Midwestern United States of America; (2) classify available research on the basis of BMPs and their efficacy in reducing P loss; and (3) highlight gaps where additional research is necessary for managing P losses with changing cropping systems.

## 2. Phosphorus Cycling and Fertilizer Recommendations

Phosphorus management requires a thorough understanding of P cycling including different biogeochemical forms of P, P storage or release from the soil, and hydrology of the landscape affecting water, runoff, and transport mechanisms (Figure 1) [13]. Historically, the goal of P fertilization to row crops was to reduce the limitation caused by P fertility issues to crop production [7]. As a result, the long-term crop rotation trials at the Morrow fields in Champaign, Illinois as well as at the Southern Illinois University Belleville Research Center in Bellville, Illinois reported to have increased soil test P (STP) [17,18]. In Missouri, 111 years of P fertilizer applications on the Sanborn field resulted in P accumulation in soil due to over-application and unrealistically high yield goals for the cropping systems evaluated [19]. Therefore, it is known that continuous additions of higher P inputs than what is exported in grain or silage will generally result in a buildup of P in soils. Another cause of the higher buildup of P in soil is outdated P recommendations to farmers [20]. Due to the cost and time involved in soil test P calibration-crop response trials, P recommendations have not been historically updated for several decades in the Midwestern USA [20]. Therefore, there is room for updating fertilizer P recommendations that can help minimize P build-up from crop production systems.

### 2.1. 4Rs of Phosphorus Management

Using the right source, rate, placement, and timing (4Rs) are critical for managing P losses from row crops. Several studies have examined the 4Rs of P management either collectively or individually [21–29]. Table 1 classifies research articles based on P application source, placement, rate, and time. In Iowa, results from 26 site-years of data from long and short-term P trials indicated that P increased yield in soils with low STP values ($\leq$16 mg kg$^{-1}$) and no response to P placement was detected at any of the optimum or high STP ($\geq$20 mg kg$^{-1}$) sites [23]. In Indiana, corn grain yield did not respond to P placement methods including deep banding, broadcast application [30]. However, Fernández and White [29] reported in Illinois that strip-till deep banded P fertilizer application produced greater kernels per row of corn, and 7.8% and 7.9% greater corn yield than no-till broadcast and shallow banded P applications, respectively. In Illinois, deep banding P increased STP levels beneath the row and lowered STP surface values compared to broadcast applications in soybean [31,32]. In the same study, authors reported that strip-tillage with deep-banded P resulted in better P uptake by soybean compared to no-till broadcast and deep banded P placement methods. The consensus among studies was that a yield response of corn or soybean was observed when STP values were very low to low (9–16 mg kg$^{-1}$) and a yield increase to deep placement of P was seldom observed, with a few exceptions [22,24,25,33].

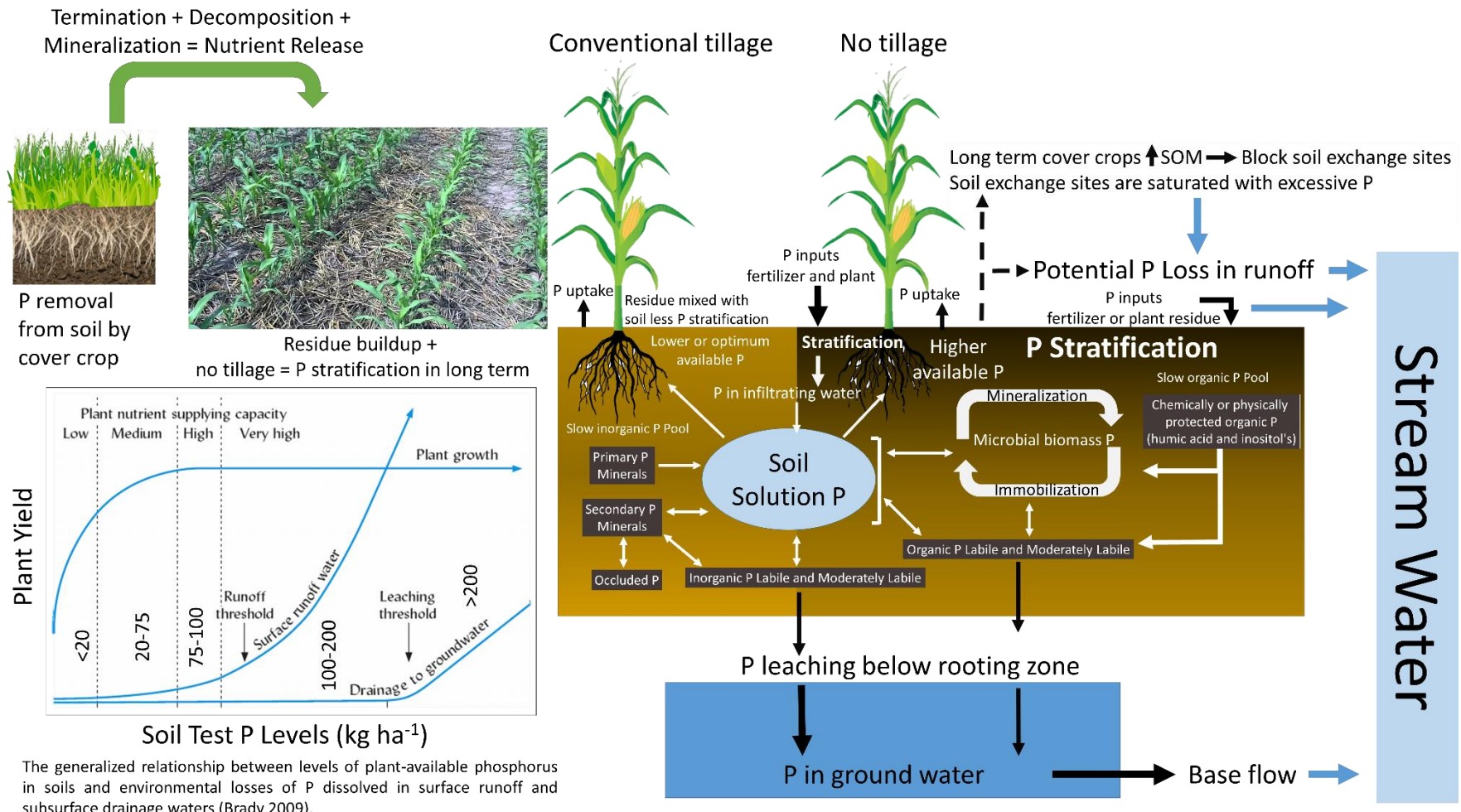

**Figure 1.** A conceptual model of P dynamics.

Phosphorus fertilizer application timing and source studies in Illinois have investigated diammonium-phosphate (DAP) or monoammonium-phosphate (MAP) sources and the interaction with application timing (fall or spring) [34]. Spring applications of P resulted in a greater yield response for corn (0.37 to 0.71 Mg ha$^{-1}$) than fall applications [34]. In Iowa, P application for no-till corn and soybean was evaluated at 20 sites in three years for a fall and spring broadcast application of triple superphosphate (TSP) on soils with STP values ranging between 6 to 29 mg P kg$^{-1}$ [35]. Yield response to P fertilizer application was observed for soils with STP values ≤21 mg P kg$^{-1}$; however, the time of broadcast P fertilizer application had no effect on no-till crop yields [35].

## 2.2. Phosphorus Stratification

To develop long-term economically and environmentally sound P management strategies, it is important to know about STP buildup with continued fertilizer applications, a decline in STP in the absence of P fertilizer application, and the critical soil test P level needed for a yield response (Table 2). Long-term study on P buildup for 8 yrs and then P decline for 36 yrs have reported corn and soybean to be at maximum yield potential when STP was >20 mg kg$^{-1}$ [36]. In Minnesota, STP buildup rates during a 12-yr continuous P application study were estimated between 0.42 to 2.49 mg P kg$^{-1}$ yr$^{-1}$ at P application rates of 24 and 49 kg P ha$^{-1}$ [37]. In the same study, the decline of STP during an 8-yr residual period (no P fertilizer applied) was estimated to be 3.3 and 0.4 mg P kg$^{-1}$ yr$^{-1}$ for initial STP levels of 40 and 10 mg P kg$^{-1}$, respectively. Crop rotation effect on P stratification needs to be researched in the production systems including winter cover crops to provide improved P fertilizer recommendations to farmers implementing these systems.

Conservation tillage practices such as no-till along with higher rates of applied broadcast P fertilizer have resulted in vertical stratification of P at the soil surface [25,28,33,38–40]. To address the issue of P stratification, alternative P placement techniques for broadcast applications including deep banding have been developed to reduce or minimize the environmental risk of dissolved P losses in surface water runoff [41–43]. In addition to limiting P losses in the runoff, the subsurface placement of P might be beneficial in enhancing P availability to crops. For example, subsurface soil layers have greater water content which can increase P uptake by roots when P was banded into the soil profile [23–25,32,33]. In Illinois, Farmaha et al. [31] reported higher protein and oil yields of soybean with strip-tillage deep banded P compared to the no-till deep band and no-till surface broadcast P because of greater soil water availability in strip-tillage deep-banded P treatments. However, continuous deep banding of P fertilizer can result in a subsequent repeating pattern of high and low STP values that can develop across fields which might result in variability of P fertility across a field and can limit our ability to accurately assess P fertility status of a field [39]. Long-term studies on P stratification linked to the impairment of water quality are limited. In addition, studies on adsorption and desorption mechanisms regulating the stratification and release of P from different pools are not available.

## 2.3. Amendments, Enhanced Phosphorus Fertilizer, and Additives

Fertilizer amendments such as lime, gypsum, alum, steel furnace slag, and ferrous sulfate biochar, as well as products like Avail (Specialty Fertilizer Products, Leawood, KS) and $P_2O_5$-Max (P-Max, Rosen's Inc., Fairmont, Minnesota, USA) have been used for managing the environmental loss of P or for increasing efficiency of applied P fertilizer [44–50]. Gypsum and lime are among the most studied amendments added to the soil for managing P (Table 3). Gypsum generally contains around 23.3% Ca and 18.6% S. Application of gypsum increases Ca levels in the soil solution and reduces soluble P loading from soil to surface waters by absorbing available P in soil solution with the Ca in gypsum [49,51–53]. Greenhouse and controlled environment experiments have reported a reduction of P loss by liming depended on initial STP status. Higher soil solution P concentrations with high STP may affect precipitation or co-sorption of Ca and P following liming. Higher pH as a result of liming decreases soil surface charges with increased pH causing P desorption that counters the

increased sorption or co-precipitation with Ca [44,46,54]. During our literature review, we did not find any long-term field-scale studies evaluating the effectiveness of gypsum and lime amendments in reducing P loss to surface and subsurface waters. Therefore, research evaluating the impact of lime and gypsum application on surface and subsurface loss of P under commercial agricultural field conditions is necessary.

The P enhancers, Avail® (Specialty Fertilizer Products, Leawood, KS) and $P_2O_5$-Max (P-Max, Rosen's Inc., Fairmont, MN), did not affect plant population, silage dry weights, grain moisture, grain yield, grain protein, grain oil, or grain starch concentrations in Missouri [47,55]. The P enhancers can increase P use efficiency and result in better nutrient management [47]. The non-response of crop yields to enhanced efficiency P fertilizers could be due to high STP of the sites selected [48]. In a meta-analysis of 503 field observations on Avail P fertilizer (maleic-itaconic copolymer acid marketed to enhance P fertilizers) applied to different crops, Hopkins et al. [48] reported an average yield increase of 2.1% while using Avail in soils testing low in STP resulted in 4.6% increase in crop yields. Slow-release P fertilizers, such as struvite and P-exchange layered double hydroxides (LDH) have been developed by recycling P from wastewater streams [56,57]. In simulated runoff studies, P loss from struvite and LDH was 1.9% and 2.4% of the applied P, respectively, whereas P loss from MAP was 42% of the applied P [57]. Slow-release P fertilizers have the potential to reduce the environmental loss of P and require further evaluation in the crop production systems.

## 3. Managing P Loss from the Surface and Sub-Surface Water Flow

Several edge of field and infield BMPs have been identified for managing P loss [10,16,58] at the source, in the transport phase, or at the sink. Figure 2 highlights different P management practices that can be implemented to reduce P loss. The effects of BMPs on P retention in the field is highly variable and largely depends on the efficiency of a BMP in removing P, time period for which a BMPs is implemented, maintenance of a BMP, the targeted area of BMP, and most importantly soil P buildup due to higher fertilizer application amounts of P to soil also known as legacy P. The BMPs to manage surface and subsurface P loss to water bodies are discussed in the following sections.

| Source | Transport | Sink |
|---|---|---|
| 1. Balance P inputs and outputs based on soil test P and yield. | 1. Minimize erosion, runoff and leaching. | 1. Develop drainage water recycling systems. |
| 2. Proper P application placement (broadcast vs deep banding). | 2. Reduce connectivity to stream. | 2. Increase residence time of P in tailwaters and ponds. |
| 3. P application timing (fall vs. spring; avoid P application on frozen ground or prior to rainfall). | 3. Implement land improvement (terraces and water and sediment retention basins). | 3. Develop constructed wetlands. |
| 4. P rate (calibrate P spreaders and update P fertilizer recommendations). | 4. Cover crops, grass filter strips and riparian buffers. | 4. Conserve existing wetlands and prairie potholes. |
| 5. Tillage (limit tillage or practice reduced tillage to compensate for P stratification). | 5. Strip cropping and contour farming. | 5. Remove sediment and sediment bound P from water bodies. |
| 6. Cover crop and crop residue management. | 6. Manage subsurface water/tile drainage water with control structures. | 6. Increase dissolved organic carbon inputs and aeration of water bodies. |
| 7. Identify critical zone areas of high STP for implementing BMPs | 7. Manage drainage ditches. | 7. Vegetative mining of P (phytoremediation). |
| 8. Animal manure management (biological and chemical treatment). | 8. Bioreactors with P adsorption filters | |
| 9. Manage grazing animal access to streams. | 9. Saturated buffers | |

**Figure 2.** Phosphorus management practices to minimize P loss at the source, in the transport phase of the movement, or at the sink.

### 3.1. Surface Water Management

Research studies have been reported on 4R's of P management and the impact of tillage systems on crop production and runoff P loss in the central U.S. [28,29,31,32,34,41,59–61]. Many researchers have reported dissolved reactive phosphorus (DRP) to be greater in surface water runoff under no-till systems compared to other tillage systems [41,59,62]. Algoazany et al. [60] reported that soluble P concentration of surface water runoff ranged between 270 to 572 µg $L^{-1}$ in a watershed under

corn-soybean production. Yuan et al. [61] reported that strip till-deep placement of P fertilizer reduced DRP loads 69% to 72% compared with a no-till-broadcast application of P, while showing no differences in grain yield due to application method. Fall broadcast P fertilizer application increased DRP and total P concentrations compared to deep banding P, but there were no differences between P fertilizer placement treatments for a spring application [61].

Most of the research on surface losses of P has focused on the impacts of tillage on P loss and to a lesser extent on rate, time, or placement (Tables 4 and 5). Phosphorus management studies have focused on crop growth and yields. Temporal changes in weather patterns influence phosphorus cycling (Figure 1); therefore, it is important to conduct studies year around to evaluate P loss in surface water runoff. The Measured Annual Nutrient loads from Agricultural Environments (MANAGE) database has classified peer-reviewed studies published on measured N and P loss in the U.S. The criteria for selecting N or P loss studies in the MANAGE database was that these studies were under cultivated agriculture, pasture, rangeland, or hay, having at least 0.009 ha in the area studied and measured nutrient loads from surface runoff due to natural rainfall [63]. The MANAGE database reported that 4% to 9% of applied fertilizer P was lost in surface water runoff [64].

### 3.1.1. Cover Crops and Tillage

Cover crops increase water infiltration, reduce surface water runoff, reduce sediment, and nutrient loss, improve soil structure, and increase residue cover [65–68]. Reductions in STP with the use of cover crops have been reported in many research studies [69–73] (Table 4). Villamil et al. [73] reported that inclusion of cereal rye and hairy vetch–cereal rye mix cover crops significantly reduced available P content in the soil compared to a crop rotation without a cover crop. The sampled topsoil depths had P stratification regardless of the cover crop treatments under the no-tillage system [73]. The redistribution of P to the surface with no-tillage was probably a direct result of surface-placement of crop residues that can result in accumulation of soil organic matter and microbial biomass in association with a lack of soil mixing. Grove et al. [40] also reported that stratification of STP was enhanced using cover crops on a silt loam soil (Figure 3).

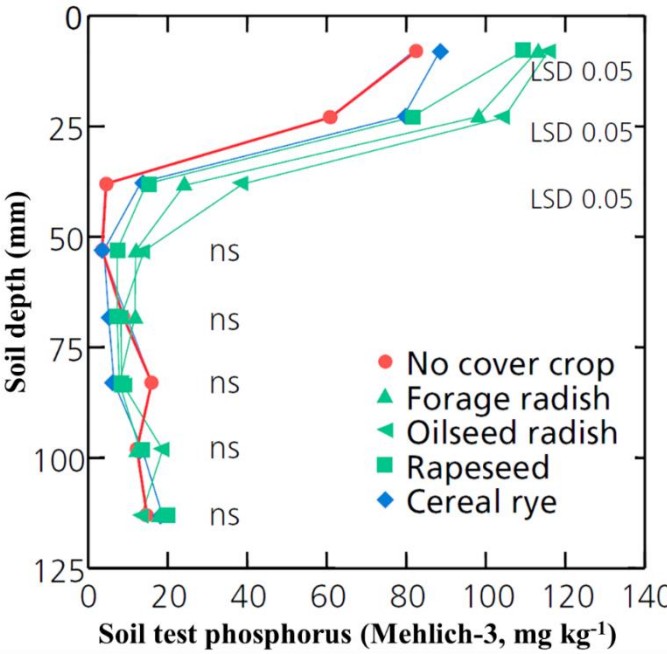

**Figure 3.** Soil test P of no-till showing the impact of cover crops on P stratification. Adopted from Grove et al. (2007).

Although reduced P loss in surface and sub-surface waters is often cited as an advantage of cover crop use [74,75], few studies have actually quantified cover crop impacts on P loss at a field or watershed scale [67,76,77]. Abel [76] reported a 52% reduction in total P losses during the first year of implementing cover crops in Kansas; however, Bruening [77] reported no reduction of total P loading in Illinois. Dissolved reactive phosphorus losses were not reduced in Kansas or Illinois after the implementation of cover crops for two years. Cover crops including weeds such as common chickweed (*Stellaria media* L.), Canada bluegrass (*Poa compressa* L.), and downy brome (*Bromus tectorum* L.) increased DRP in runoff from no-till soybean as compared to a no cover crop control [78]. In a runoff and tillage study in Minnesota, available P concentrations in surface water samples increased with an increase in residue cover [79]. In general, DRP losses increased in a no-tillage system and sediment-bound total-P increased in tilled crop production systems. Most of the studies that have evaluated cover crop impacts on P loss have been conducted as rainfall simulation studies [78,80,81]. Results from simulated rainfall studies indicated trends for P loss for field-scale conditions, but could not show losses that occur due to actual precipitation received. Therefore, cover crop and tillage effects on P loss should be studied using multi-year, field-scale, and watershed projects with natural rainfall. Cover crop effects on runoff and P loss will change throughout the year with the cover crop or grain crop growth. Field-scale data are needed to document the impact of cover crops on P loss to provide recommendations for their use as a BMP for managing P.

Topographic variation along with tillage practices in large agricultural fields can have a substantial impact on dynamics of soil P as well as the performance of cover crops [79,82–84]. Soil organic matter levels vary in response to topography. Soil organic matter is regarded as one of the main soil properties blocking iron and aluminum exchange sites of soil where available P can be attached which could result in increased organic and/or inorganic P losses [85–88]. Phosphorus loss due to the interaction between topography, tillage, and cover crops needs more exploration. Temporary immobilization of P in the cover crop biomass along with a surface cover of biomass may improve runoff water quality by decreasing sediment loss and sediment-bound P. However, P stratification in topsoil over long period of time due to higher biomass inputs of cover crops may act as a source of P resulting in greater DRP losses. Stratification of P in inorganic (soluble or loosely bound P, aluminum bound P, iron-bound P, reductant soluble P, calcium bound P) and organic (labile organic P, moderately labile organic P, humic and fulvic acid P) pools of P in soil have not been reported by any study under different cover crops and tillage systems.

### 3.1.2. Water and Sediment Control Basins (WASCoBs) and Terraces

Land improvements for soil conservation have been promoted in many Midwestern states of the U.S. Common conservation practices have cost-share programs in several states include installing terraces and WASCoBs. Water and sediment control basins are constructed along minor slopes or watercourses in order to capture runoff during storm events and then slowly discharge the runoff water through a stable outlet. The WASCoB reduces erosion and improves water quality by removing the total suspended solids and P load in runoff [89]. Research on terraces and WASCoBs has focused on the management of drainage water for erosion control (Table 5). Therefore, research on the benefits of improving water quality and managing nutrient loss in runoff was explored to a lesser extent in these land improvement conservation practices. A 5% decrease in peak stormwater flow by using WASCoBs was reported in Iowa [89]. Total suspended solids in surface water runoff were reduced up to 80% along with 85% reduction in phosphorus loads to surface waters with the use of WASCoBs [89]. Similarly, Edwards et al. [90] reported a 94% reduction in sediments, a 76% reduction in N loss and a 52% reduction in P loss by establishing a basin for trapping agriculture runoff water. Fiener et al. [91] monitored WASCoBs for eight years for sediment and nutrient retention, and 54–85% of sediments were retained in the WASCoBs basin. Peak runoff and peak concentrations of agrochemicals were reduced by at least 50% with the implementation of WASCoBs [91]. The eight-year study conducted by

Minks et al. [92] showed a 260% reduction in suspended sediments by the use of sediment control structures [92].

In a comparison study of various conservation practices (no-tillage, contour farming, terraces, WASCoBs, vegetative filter strips), Czapar [93] observed that terraces and WASCoBs had the lowest soil loss (0.4 tons' acre$^{-1}$ yr$^{-1}$) and total losses of N and P in soil and water were 7.28 kg N ha$^{-1}$ and 1.8 kg P ha$^{-1}$, respectively, compared to other conservation practices [93]. Terraces and WASCoBs were reported to reduce total-P loads and concentrations by 5.8 kg P ha$^{-1}$ and 1.4 mg L$^{-1}$, respectively, when compared to runoff water before and after installation [94,95]. However, it is important to point out that the dissolved fractions of N and P were not reduced using terraces or WASCoBs [95]. Using a modeling approach for tile-drained terraces, Gassman et al. [96] found an 80% reduction in sediment-bound P (APEX model), and 64% to 74% reduction of sediment and organic P, using the SWAT model.

In Nebraska, Mielke [97] studied the performance of WASCoBs for erosion control and drainage water management. Sediment trapping efficiency of WASCoBs exceeded 97% and the sediments that were discharged from the outlet had 12% silt and 88% clay in suspension after a two-hr runoff event measured at the outlets [97]. The authors of this study pointed out that the sediment trapping efficiency of the WASCoBs can be decreased by more than half if the newly built WASCoBs were not protected from erosion following construction. Therefore, it is important to tie infield BMPs focused on the surface cover, such as cover crops, with newly built terraces and WASCoBs to extend their life. This combination of multiple BMPs can also help nutrient retention in agriculture fields (Figure 4). Overall, there is a need to integrate the right (4Rs) fertilizer management systems along with other conservation strategies like cover crops, terraces, WASCoBs, saturated buffers, and bioreactors acting as staged water management systems to demonstrate that BMPs working in concert can systematically minimize nutrient loss (Figure 4).

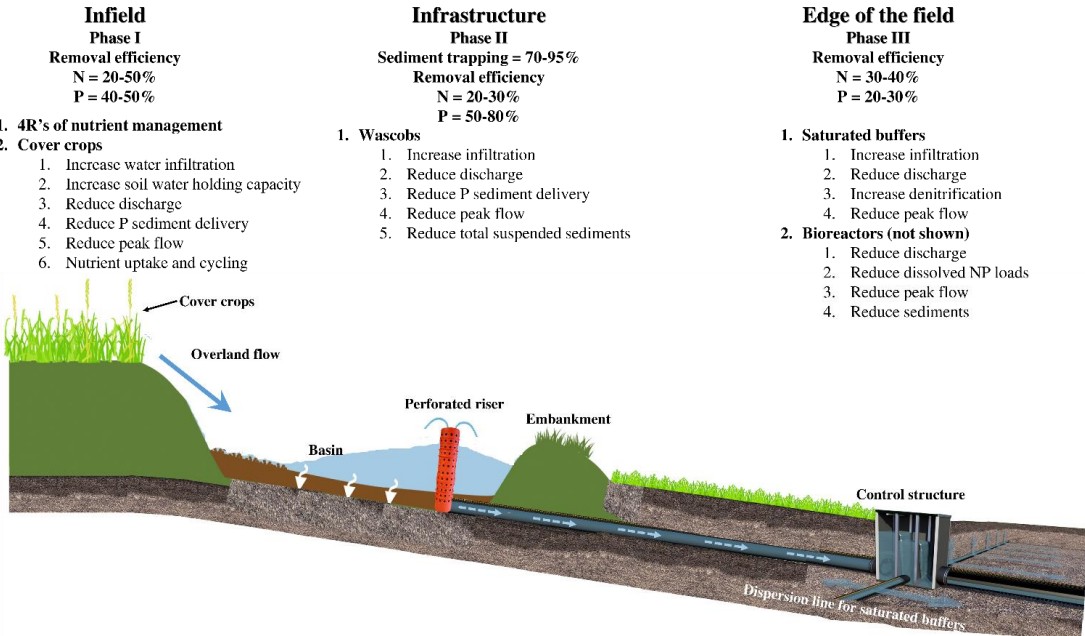

**Figure 4.** An example of an integrated best management system for an agriculture field with cover crops implemented in phase 1 followed by water and sediment control basins (WASCoBs) in phase 2, and the edge of field practices such as saturated buffers in phase 3. Nutrient use efficiencies listed in the figure is a general range taken from several publications given in this review.

Construction of terraces result in manmade topographic positions, which can be classified into the shoulder, backslope, footslope, and channel [98]. The change in the landscape can substantially influence crop yield and increased input of fertilizer amendments may be necessary to meet the production goals. In Missouri, channel slope compared to other landscape positions of the terraces

showed a significant reduction in corn and soybean yields due to abiotic stress to the plants caused by waterlogging [98]. Research on supplementing channel slopes with subsurface tiles to improve drainage and changing the design of inlets to a blind inlet packed with steel slag or industrial material that removes P should be evaluated [16]. In a 7-year paired comparison of open and blind inlets of tile drains in Indiana, total P and DRP loads were 66% and 50% lower, respectively, from blind inlets than from open inlets [99]. In a similar study conducted in Minnesota, the conversion of an open inlet to blind inlets resulted in a decline in median total suspended solid concentrations from 97 to 8.3 mg L$^{-1}$ and dissolved P concentrations from 0.099 and 0.064 mg L$^{-1}$, respectively [99].

Terraces and WASCoBs with open and blind input of the tile lines must be evaluated for long-term profitability by establishing a cost for the reduction of P loading that occurs using these land improvement practices. It is important to point out that if there is no cost-share program to support these practices it could add substantial construction costs [16,95]. The P abatement costs for WASCoBs based on four studies was estimated by Liu et al. [16] and ranged between \$1.88 to \$1065.93 kg$^{-1}$ P removed [93,94,100,101].

### 3.1.3. Vegetative Buffers

Vegetated buffer strips (VBS) consist of non-cultivated borders between surface waters and agricultural fields to improve water quality and biodiversity [102]. The main mechanisms for P retention in VBS are sediment deposition, water infiltration, nutrient adsorption, and plant absorption [103]. Dense aboveground vegetation and belowground root systems in a VBS first remove P from overland flow by deposition and infiltration. Aboveground vegetation in the VBS reduces flow velocity of surface water, available energy for particulate transfer and increases hydraulic roughness [104,105], whereas below-ground root systems increase soil permeability and porosity while increasing infiltration of overland water flow [106,107].

The effectiveness of VBS depends upon the vegetation characteristics (single species vs. multiple species, different crop or tree species, and plant density) [103,108], as well as width and slope before the surface water flow reaches buffer area [109]. For example, Abu-Zreig et al. [103] reported that the P removal efficiency of a VBS ranged from 31% in a 2-m VBS to 89% in a 15-m VBS. Abu-Zreig et al. [103] concluded that VBS width was the most important factor determining P trapping compared to other factors including the rate of inflow, type, and density of vegetation. A study conducted by Lyons et al. [110] found that forested riparian buffers were less effective than the grass riparian area in trapping total suspended sediments.

Vegetated buffers generally retain more particulate P than dissolved P from overland flow [102,111]. In a literature review on P retention of riparian buffers, Hoffmann et al. [111] reported that retention of total P and DRP by riparian buffers ranged between 41% and 95% and −71% to 95%, respectively. Stutter et al. [112] found increased inorganic P solubility indices, dissolved organic P, phosphatase enzyme activity, microbial diversity, and biomass P in VBS soils compared to soil in an adjacent field. Roberts et al. [102] concluded that the establishment of VBS increased labile organic and inorganic soil P fractions from previously immobilized soil P from fertilizer applications or sediment P from runoff by increasing soil P cycling rates and potentially increasing the amount of P available for leaching to water in streams.

In a literature review on the effectiveness of current conservation practices on P loss, Dodd and Sharpley [113] found that labile soil P forms accumulated over time in the buffer zones, including vegetative and grass filter strips, became the source of both organic and inorganic dissolved P. Remobilization of P assimilated by biological processes as from particulates to soluble P forms can occur by microbial turnover, decrease in microbial biomass, drying and rewetting cycles, and leaf senescence which can increase water-extractable P at the soil surface of VBS and increase the risk of dissolved P loss Roberts et al. [102]. However, Roberts et al. [102] mentioned that the increase in dissolved P from VBS was based on limited evidence and needed further investigation. Roberts et al. [102] concluded that VBS became a modified pathway in the transfer of P to surface water streams

by altering the timing, extent and chemical form of the P that was delivered to streams, which could be due to differences in biological activity between the agricultural field and VBSs. Therefore, the VBS needs to be properly managed to improve P retention of all chemical forms including particulate P as well as soluble P forms from overland flow.

Additional research will help us to understand the interaction of vegetated buffers with different management practices including tillage, fertilizer applications, soil liming, and residue management in the agricultural fields. Research finding a balance between P uptake efficiency, P removal from the soil, and rate of biomass harvest from VBS should be supplemented with research on providing recommended actions for maintaining existing VBS. Designing VBS with an engineered P management structure having a P adsorption material to treat dissolved P and identifying plant species that increase P uptake from engineered structures and soil can be a potential option to reduce dissolved P losses.

*3.2. Subsurface Water Management*

The 2012 Agriculture Census reported that 19.65 million ha in the US had subsurface tile drainage [114,115]. In Illinois, around 35% of the 10 million ha of crop production area was tile-drained [116]. There are a few studies that have monitored P losses from tile drains. Hydrological modification of the agriculture landscape with tile drainage to target higher production goals has led to a decrease in the residence time of water in the soil profile. A long-term study on tile-drained water quality in the Little Vermilion River (LVR) watershed, IL, has provided results about DRP and total P losses [60,117]. Dissolved reactive phosphorus is usually high during precipitation events that followed the widespread application of P fertilizer on frozen soils [117]. In a 50-year analysis of published data on P loss in drainage water, Christianson et al. [118] reported that less than 2% of applied P was lost in drainage water and no-tillage significantly increased DRP loads in drainage water flow compared with conventional tillage. Phosphorus loading to subsurface tiles can be impacted by preferential flow in soil, soil P sorption capacity, soil redox conditions, soil-test P levels, tillage, cropping system, source, rate, placement, and timing of P application, drainage design/installation, and spatial/temporal variation and precipitation [119]. Additional research is needed on tile drain spacing, depth, and layout effects on P loss.

Several conservation practices like drainage water flow control structures, bioreactors, and saturated buffers have been promoted in the Midwestern U.S. to manage and connect tile drain water to the riparian buffer zone (Table 6). Some of these conservation practices are being researched for their feasibility; however, the question remains is how much water can be treated with these conservation practices. Additionally, what size of a precipitation event can be managed with these practices and how do these practices work in concert with other conservation practices like nutrient management planning, cover crops, and land improvement practices.

3.2.1. Controlled Drainage

Controlled drainage structures also known as inline water level control structures, can regulate subsurface tile drainage flow (Agri Drain Corp., Adair, IA). These structures, if actively operated, can efficiently manage the water table of the field [120–122]. Drainage water control structures have been shown to reduce N and P loading to streams by reducing the amount of discharge and increasing uptake of these nutrients by cash crops [120–122]. However, concern has existed over the potential for greater dissolved P losses with controlled drainage due to the reductive dissolution of P bonded to iron during periods of water stagnation. Based on a laboratory study, Valero et al. [123] concluded that elevated water tables caused by drainage water management could increase P export in subsurface drainage following the reductive dissolution of iron-bound P in waterlogged soils.

In contrast to Valero et al. [123], field trials in Missouri have documented some significant benefits of reduced P loading [121,122] and increased corn and soybean production with controlled drainage [124,125]. In a 4-year study, controlled drainage was reported to reduce flow-weighted DRP concentrations by 0.06 mg L$^{-1}$ compared to free drainage [121]. Reduction in P loading by using

control structures was due to a 63% reduction in annual drainage and increased P uptake by corn from water during the dry summer months, which further reduces the amount of P available for leaching to drainage tiles in the following spring [121]. Zhang et al. [126] also reported that controlled drainage with sub-irrigation was an effective way to reduce annual and cumulative losses of DRP, particulate P, and total P in tile drainage when compared to free drainage. Innovative techniques like P removal filters (sand enhanced with iron) can be connected to subsurface tiles and control drainage structures to further reduce P loading into streams [127].

Decisions for regulating the water table of a field can be input into a model to give recommendations on when to use stop logs in controlled drainage structures and when to release water from the fields to prepare them for planting. Several models have the potential to be used for modeling P losses in artificially drained fields [128]. Several components of these models need to be assessed before making the simulation. Phosphorus modeling components include partitioning of water and P into a runoff, macropore flow, and matrix flow. Once in the soil matrix, sorption and desorption of dissolved P, filtering of particulate P, and inputs–outputs of P need to be considered in the models. Multi-location and long-term field experiments are required for developing the models for managing artificially drained fields with control structures and reducing P losses from these drainage systems.

### 3.2.2. Bioreactors

Bioreactors are the edge of field BMP that intercepts the drainage water from tile drains and removes excess nutrients from this drainage water before releasing it to surface water bodies. Bioreactors are traditionally designed for reducing N loading from drainage water; however, modifications in the bioreactor design have been evaluated for their feasibility in reducing P loss [129–132]. Phosphorus sorbing materials in the bioreactors should have high P bonding ability even at low P concentrations in the drainage water and sufficient hydraulic conductivity for high flow conditions [130]. Generally, P sorption ability is negatively correlated to hydraulic conductivity among P sorption materials [130]. A 65% reduction in P concentration was observed for laboratory-scale bioreactors with biochar [129]. Hua et al. [133] reported that a recycled steel by-product filter effectively removed phosphates from the effluent and the total phosphate adsorption capacity was 3.70 mg P $g^{-1}$ under continuous flow conditions, compared to woodchip bioreactors. In addition to the removal of dissolved P, woodchip bioreactors coupled with a solid settling tank have the potential to remove particulate P from the tile drain water [134].

Field studies of bioreactors designed to treat N and P from tile drainage water have shown a reduction in dissolved and total P loading [135,136]. Husk et al. [136] evaluated bioreactors filled with a mixed media reactive to P for three years and found a 19 times higher reduction in total P loads with mixed-media bioreactors compared to a standard woodchip bioreactor. However, both types of bioreactors were not able to reduce the total P concentration below the critical environmental threshold level of 0.03 mg $L^{-1}$ [136]. In South Dakota, a woodchip bioreactor supplemented with a P adsorption structure filled with mixed-media reduced dissolved P 10% to 90%, and P removal rates varied between 2.2 and 183.7 g $m^{-3}$ $day^{-1}$ during the study period [135].

Design criteria of bioreactors used for abatement of N and P should consider factors including the drained area, retention time of water needed for biochemical processes in the bioreactor, percent of flow that can be treated, stormflow and baseflow of waters during and after precipitation events, operating failure due to sedimentation, operating temperature, solution pH regulating sorption processes of P, life expectancy of filter material, and cost of N and P removal. Several engineered materials are being developed to remove P from effluent water including reduced graphene oxide membranes/filters, nanocrystalline zinc-iron layered double hydroxides and other metal hydroxides [2,137,138] which have P adsorption capacities as high as 140 mg P $g^{-1}$ of material. Along with research needed on the design criteria of the bioreactors, research efforts need to be directed towards evaluating the engineered material at the field scale as well as testing bioreactors with other crop management practices.

### 3.2.3. Saturated Buffers

Vegetative buffers at the edge of the field with natural or planted vegetation have the potential to reduce sediment loss, nutrient transport to surface waters, and reduce N and P from shallow groundwater. In tile-drained areas of the Midwest, vegetative buffer areas are not effective because the tile lines bypass the buffer and discharge directly into streams or ditches. A practice that has shown to be a cost-effective method to remove $NO_3$-N and dissolved P from tile drain water is saturated riparian buffers [139–141]. Saturated riparian buffers are used in situations where a field is bordered by a vegetated buffer, typically along a waterway or stream, and drained by a subsurface drain tile network. A saturated buffer is incorporated into an existing or newly established riparian buffer in which a shallow lateral line intercepts existing tile lines and disperses the water across the vegetated buffer. As the water flow is dispersed across the buffer, there is a high potential to reduce nutrient concentrations and loads by direct uptake of the vegetation in the buffer or through the conversion of $NO_3$-N into nitrogen gas and for DRP adsorption to iron and aluminum exchange sites in the saturated buffer area. Drainage outflows are also reduced along with a reduction of the nutrient load because a portion of the water dispersed across the buffer is taken up and transpired by the vegetation. Overall, this practice requires a relatively low installation investment and very little management or maintenance costs.

The potential of saturated buffers for reducing P losses has been explored to a lesser extent [142]. Utt et al. [142] concluded that saturated buffers cannot appropriately treat P related water quality concerns since saturated buffers monitored for a reduction in dissolved P loss from the tile water showed no consistent trends. However, modification of saturated buffer designs including backfilling saturated buffer areas and tile lines with steel slag or industrial by-products of higher ion exchange potential should be explored [143]. McDowell et al. [143] used by-products from steel and energy industries to mitigate P loss from tile drains and reported that dissolved P and total P loads in tile drain were 0.27 and 1.07 kg P ha$^{-1}$ lower than the control. The retention time of the water in the saturated buffer should be increased and industrial by-products/material that show promising results in reducing P loss in water can be engineered in the design of the saturated buffers. The adsorption and desorption isotherms of the material used in the saturated buffer design could be tested prior to their implementation (Figure 5).

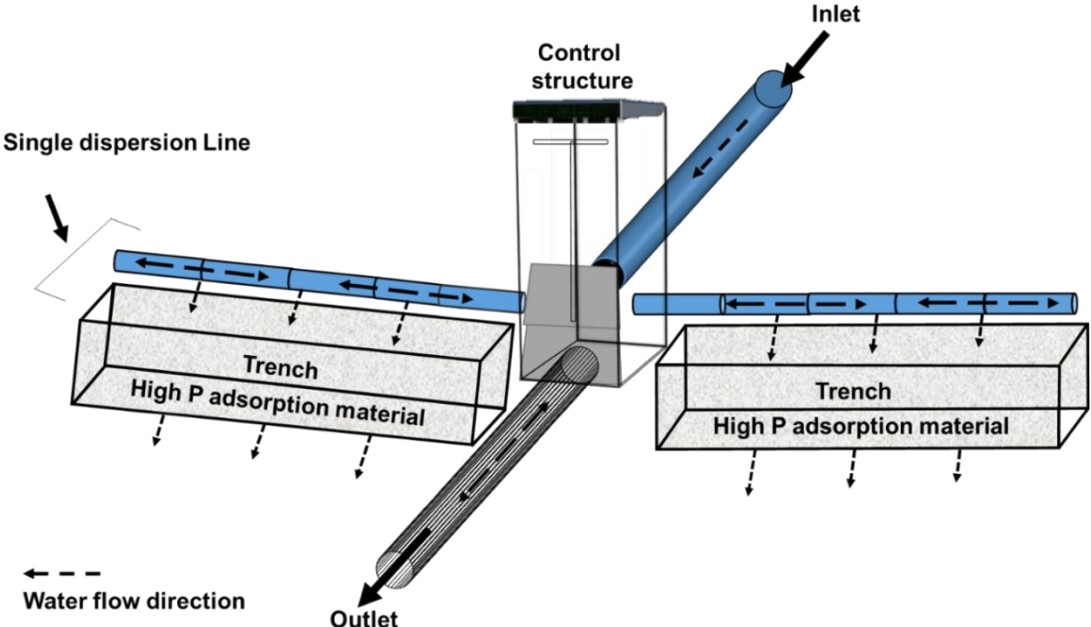

**Figure 5.** Design of the saturated buffer for P removal using a high P adsorption material.

### 3.2.4. Constructed Wetlands, Reservoirs, and Drainage Water Recycling

Constructed wetlands are strategically located in the agriculture landscape to store water, nutrients, and particulates for both short and long-term periods (Table 7) [144]. Research on constructed wetlands has shown 27% to 100% retention of P [145,146]. Constructed wetlands can store sediments as well as can serve as a sink for total and dissolved P; however, seasonal differences in P retention can be significant with higher retention of P during summer than in winter [145]. In Georgia, the water quality of inflow and outflow of a restored wetland was monitored for 8 years with a 66% reduction in dissolved P and total P loss [146]. Total P removal in the wetlands of the Des Plaines river wetlands demonstration project near Chicago, Illinois, USA ranged from 52% to 99% [147]. A wetland designed with a grassed buffer to remove sediments and total P from surface runoff removed 88% of 48 kg ha$^{-1}$ total P load during a 2-yr study period [148]. In contrast, some of the constructed wetlands in Illinois and Maryland did not show a significant reduction in P [149–151]. A lack of P reduction in these studies was attributed to variation in seasonal precipitation contributing to runoff and discharge to these wetlands, and insufficient wetland acreage for sufficient P-binding/removal. Research on constructed wetlands is needed for developing mapping tools for strategic planning and layout of wetlands in agricultural watersheds and validating mapping tools and models with measured data. Additionally, research evaluating amendments like calcium that can be supplemented to increase P adsorption in wetlands, a life expectancy of constructed wetlands, and the economic benefit of P removal compared to construction cost should be evaluated in long-term studies [152].

Reservoirs and ponds at the farm scale serve an important role in reducing P loading into streams and rivers. Phosphorus removal by reservoirs or ponds depends on several factors including the concentration of soluble P in water discharged, concentration of soluble P in the overlying water column of the pond, hydraulic residence time of P in the pond, pH of pond water, concentration of P in underlying sediment, adsorption and desorption of P by pond sediments, weather parameters including light and temperature governing biochemical reactions, P uptake and assimilation by hydrophytes, and dissolved N and carbon in the water [153]. Reddy and Reddy [154] reported a longer hydraulic residence time for water was needed in ponds for P removal at high P loading compared to low P loading. In Minnesota, ponds supplemented with different species of duckweed (*Lemnaceae*) affected P absorption differently [155]. *Lemna minor* L. and *Spirodela polyrhiza* (L.) Schleid. were the most stable nutrient sinks and removed the largest amount of P from ponds during the 8 week study period [155]. On-farm stormwater retention ponds can serve as a sink for nutrients, enhance groundwater recharge, delay flooding potential, increase on-farm evaporation, and most importantly serve as a source of water to grow crops during drought years. Drainage water recycling research from farm ponds for crop production is limited. Long-term research on drainage water recycling is needed to demonstrate the economic benefit in terms of greater crop yields [124,125] and improved quality of discharged water from this practice [43,156].

## 4. Watershed Scale Studies and Critical Source Area Concept

Several catchment or watershed scale studies have evaluated P risk assessment after implementing BMPs [67,157,158]. Lemke et al. [158] reported no significant reduction in P loss in tile-drained sub-watersheds (4000 ha) of the Mackinaw River in Illinois after 7 years of implementing BMPs. The authors concluded that BMPs established during the study were not adequate to manage nutrient export from subsurface drainage tiles. Singh et al. [67] reported a significant reduction in discharge and total soluble solids by implementing cover crops for two years during the treatment period of headwater agricultural watersheds (<50 ha). No improvement in event mean Nitrate-N, ammonium-N, and dissolved P concentrations in stream water quality was reported [67]. Event means of dissolved P losses were increased by 60% during the cover crop treatment period. At the watershed scale, losses of legacy P can contribute to a possible lag time in response to restoration treatments or management systems [159]. Due to variability induced by several factors including hydrologic processes, landscape position, soil series, soil P status, crop P uptake, tillage, and crop rotations at the watershed scale, these

studies need to be replicated or designed using a paired watershed approach to detect the impact of BMPs for improving water quality.

The critical source area of P export from agriculture watersheds can be defined as the area where a high concentration of soil P (source area) coincides with areas that have a high potential of runoff or subsurface flow of water (transport area) [160]. Many studies have found that large runoff amounts were contributed by less than 10% of the watershed area [161–163] and a significant part of the watershed had little impact on nutrient transport and loading to headwater streams. Therefore, critical areas can be targeted with the implementation of BMPs. Identifying areas in a field with high P status can be done with sampling for STP along with developing spatial P maps of the fields in watersheds using soil electric conductivity or ground penetrating radar models (Geonics, ON Canada or Veris, OR USA). High-resolution digital elevation models can be used for developing soil drainage class, topographic position index [164], topographic wetness index [165], and the network index [166] for identifying areas in the watershed that have surface hydrological connectivity. Additionally, soils that have a restrictive layer like a fragipan can be identified using ground sensing radars and soil electric conductivity to develop spatial maps of subsurface hydrological connectivity, which can contribute to increased P loading [161]. Soil P and surface/subsurface layers can be used to identify critical source areas (Figure 6). Weld et al. [167] evaluated the relationship between soil P and surface runoff P using the P index for identifying critical source areas. They reported that areas vulnerable to P loss were located along the stream channels where areas of runoff generation and areas of high soil P coexisted. In Illinois, Evans [163] developed an algorithm to identify critical source areas using topographic position index and soil test P and reported that critical source areas in row-cropped watersheds occurred near grass waterways and roadside drainage ditches. Research on identifying critical source areas in agricultural watersheds and sampling for P loss is needed to further validate the models predicting the critical source areas. The identified critical source areas can be targeted with BMPs and monitored for P reduction over time.

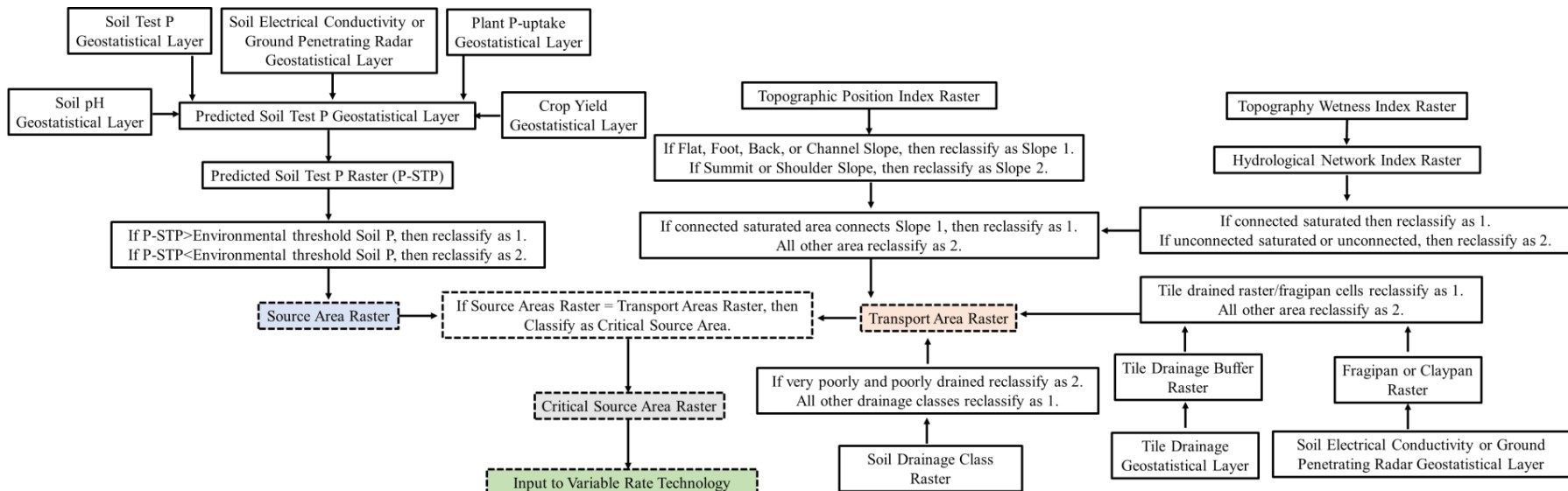

**Figure 6.** Flow chart for identifying critical source areas.

## 5. Phosphorus Index

The phosphorus index was introduced for the first time as a nutrient management tool in 1992 and modified versions have been adopted by many states in the US [168,169]. The intended goals of the P Index were to: assess the risk of P transport from fields to water bodies; identify critical parameters influencing P loss, and help to identify BMPs that would decrease P loss to water bodies. In the last 25 years, a considerable amount of research has been conducted on refining the P index [5]. Many of the P indices have been evaluated for their potential to predict site vulnerability to P loss by comparing simulated model outputs and validating in the field with measured data [170–172]. Nelson and Shober [172] reported that future research on P indexes should focus on (i) P Index evaluation, (ii) advancement of P indices, and (iii) interpretation and implementation of P Index results.

For example, the Illinois P index is an additive P index and was updated in 2013 to include site characteristics and source factors as two important components of the P index approach [173]. Due to legacy P inputs, a significant improvement in water quality has not been reported [6], which has raised concerns about the continued use of the P index approach. At this point, Illinois does not have any research evaluating the Illinois P index and research from P indexes in other states in the US have reported mixed results [170,171,174]. Therefore, research identifying parameters to be included in the P index of Illinois that can accurately assess the risk of P loss is needed. Additionally, there is a critical need to evaluate the Illinois P index and compare it to other P indices to determine if the Illinois P index appropriately identifies the impact of soil, climate, and management practices on P loss. Other states may have similar issues that need research to validate nutrient management tools.

## 6. Gap Analysis and Conclusion

Phosphorus application rate recommendations for corn–soybean are typically based on yield response to STP levels and no component of the environmental threshold for P loss is included in this recommendation. However, there are several potential areas where adopting BMPs that show a reduction in P loss can be implemented. The first step would be to target high P buildup areas where updated P fertilizer recommendations to farmers could be provided. Updating P fertilizer recommendations can result in direct benefits of minimizing P losses from crop production systems. Long-term studies on P stratification linked to the impairment of water quality and adsorption–desorption mechanisms regulating the stratification and release of P from different pools of soil to water are limited. To include an environmental threshold P loss potential of the fields, multi-location and long-term field experiments are required for developing the modeling approach. The modeling approach can identify critical source areas in the fields and can help in developing site-specific P fertilizer recommendations. Additionally, research on developing innovative, inexpensive, reliable, and rapid methods for estimating STP levels is needed that can be used in nutrient management stewardship.

Field-scale studies evaluating the effectiveness of lime, gypsum application, and slow-release P fertilizers on surface and subsurface loss of P under commercial agricultural field conditions are needed. Research on P source, rate, placement, and time can be further evaluated for their impact on managing P loss in surface water runoff and tile drain water. Simulated runoff studies are common and are a faster method to test BMPs recommended for managing P loss; however, the temporal changes in weather patterns govern P cycling. Therefore, year around, natural rainfall studies evaluating P loss in runoff water should be supported. Comprehensive cover crop and tillage effects on P loss must be studied using multi-year, field-scale and watershed projects because the effects, cover crops have on runoff and P loss will change throughout the year as a function of the cover crop or cash crop growth. Soil amendments and cover crop cropping systems that enhance P sequestration and timely mineralization of P that is available for cash crop uptake should be developed.

Several factors including preferential flow in soil, soil P sorption capacity, soil redox conditions, STP levels, tillage, cropping system, 4Rs of P, drainage design and installation, spatial-temporal variation, and precipitation can impact P loading to subsurface tiles. Utilizing GIS, remote sensing,

and chemical–environmental tracers for identifying linkages of drainage water flow delivering P to headwater streams need to be addressed in future research so that models can be developed and validated in fields for managing P loss. New conservation practices including improved designs of controlled drainage, bioreactors, and saturated buffers need long-term monitoring at field and watershed scales.

An emphasis on watershed-scale studies that monitor long-term P fluxes should be supported with more funding. Since stream water quality may not respond to BMPs in the short-term, the length of the watershed scale studies needs to be considered due to legacy P impacts. Several BMPs discussed in this review can be either used singly or collectively for managing P loss. However, the efficiency of BMPs in reducing discharge and P loss is highly variable and there is a need to further explore BMPs to help facilitate the development of appropriate watershed management plans. The short- and long-term retention of P by BMPs should be evaluated and criteria to maintain and remove P from BMPs should be developed so that P is not exceeding the retention capacity of the designed BMP.

**Author Contributions:** Investigation, G.S., G.K.; data curation, G.S., G.K.; writing—original draft preparation, G.S., G.K.; writing—review and editing, G.S., G.K., K.W., J.S., K.A.N.; funding acquisition, G.S., K.W., J.S. All authors have read and agreed to the published version of the manuscript.

**Funding:** This research was funded by Illinois Nutrient Research & Education Council.

**Conflicts of Interest:** The authors declare no conflict of interest.

**Table 1.** Research papers classified based on phosphorus application source, placement, rate, and time.

| Purpose | State † | Soil Series * | Crop ** | P Source | P Placement | P Rate (kg P ha$^{-1}$) | P Timing | Tillage | STP (mg kg$^{-1}$, kg ha$^{-1}$ *) | Tile Drain | Irrigated | Study Type | Reference |
|---|---|---|---|---|---|---|---|---|---|---|---|---|---|
| P application and STP effects on P uptake and crop yield | IN | Raub SiL | C, S, W, H | SP | Mixed by disking | 7.9, 19.96, 39.9 | na | DT | 17.6, 30.8, 55 | na | na | Placement | Barber [175] |
| Deep P fertilizer application and subsoiling on a claypan soil | MO | Mexico SiL, Putnam, SiL | A | TSP | Surface or mixed with soil | 181 | na | CT | na | na | na | Placement | Jamison and Thornton [176] |
| Relative efficiency of P placement methods (broadcast vs. banded) | IL | Zanesville, Elliott, Muscatine SiL | C | SP | Broadcast, banded | 0, 4, 8,16, 6, 12, 24 | na | | 2.5–9 | na | na | Placement | Welch et al. [177] |
| Water-soluble P source and placement method effect on corn growth and P uptake | IL | Proctor SiL | C | na | Mixed, banded, with seed placement | 0, 300, 600 mg P/pot | na | na | 5.5 | na | na | Placement | Garg and Welch [178] |
| NT impact on corn yields, soil pH, soil fertility, and compaction | VA | Lodi loam | C, CC | na | na | na | na | yes | na | na | na | Tillage, cover crops | Shear and Moschler [179] |
| Combination of row and surface applied fertilizers on NT corn yields in low and high fertility systems | OH | Canfield SiL | C | TSP | Broadcast, mixed | 9, 55 | na | yes | 30* | na | na | Tillage | Triplett and Van Doren [180] |
| Determination of dominant forms of P present in poorly to somewhat poorly drained prairie soils; Quantification of vertical and horizontal P movement within the landscape | IL | Tama; Muscatin; Sable; Denny | na | na | na | na | na | na | na | na | na | P availability | Smeck and Runge [181] |
| P uptake from surface-applied fertilizer by NT corn planted in low soil P content | KY | Zanesville SiL | C | SP | Broadcast, banded near-seed | 0, 56, 112, 224 | na | NT | 3* | na | na | Placement, rate | Belcher and Ragland [182] |
| Crop response to P and K placement methods | IA | Webster SiL | C, S | SP | Broadcast, banded | 0,29,58 | na | na | na | na | na | Rate | DeMooy et al. [183] |
| Fertilizer placements impact on soybeans | MN | MeIntosh SiL, Webster SiCL, Nicollet SiCL | S | | Banded, seed placemen, broadcast | Vary | na | na | na | na | na | Placement | Ham et al. [184] |

**Table 1.** *Cont.*

| Purpose | State † | Soil Series * | Crop ** | P Source | P Placement | P Rate (kg P ha⁻¹) | P Timing | Tillage | STP (mg kg⁻¹, kg ha⁻¹ *) | Tile Drain | Irrigated | Study Type | Reference |
|---|---|---|---|---|---|---|---|---|---|---|---|---|---|
| Tillage and P placement methods impact on corn growth and yield. | IL, NE | Sharpsburg SiCL, Leshara SiL, Platte SL, Flanagan SiL | C | OA | Broadcast, banded | 0, 11, 22, 45, 30, 60 | PP | CP, MP | 4.2–12.8 | na | Both | Placement, rate, tillage | Cihacek et al. [185] |
| P fertilizer placement method impact on P utilization, N uptake and N₂ fixation | MN | Waukegan SiL | S | SP | Broadcast and incorporated, broadcast, banded near seed, banded at some distance from row | 35 | na | na | na | na | na | Placement | Ham and Caldwell [186] |
| Interaction of weather and soil variables with P fertilizer application rates, sources and methods | IA | Kenyon SiL, Readlyn, Floyd, Clarion-Webster, Primghar silty clay, Grundy SiL, Edina SiL | C | RP, SP | Broadcast, in-row | 22.5, 45, 67, 134, 268 | na | na | na | na | na | Source, rate, placement | Casanova [187] |
| Comparison of tillage systems | IA | Loess Hills, Monona-Ida-Napier soils | na | na | na | na | na | na | na | na | na | Tillage, runoff and placement | Johnson et al. [79] |
| Residual effects of different P fertilizer application rates and placement on Soil-P solubility | ND | Parshall fSL | SW | TSP | Broadcast, banded | 0, 20, 40, 80, 160 | PP | CT | 6.6 | na | Dryland | Placement | Alessi and Power [188] |
| N and P placement method impact on P uptake and crop yield | KS | Hastings Si, Cherokee SiC, Woodston SiC | WW | APP | Broadcast, knife | 0, 20 | PP | na | 4–22 | na | na | Placement | Leikam et al. [189] |
| Spacing of N-P fertilizer bands influence on crop yield and P uptake | KS | Crete SiC, Parsons Si, Pond Creek Si, Pawnee C | WW | APP | Banded | 0, 6, 12, 24 | PP | na | 4–10 | na | na | Placement, rate | Maxwell et al. [190] |
| Effect of different spacing of fertilizer placement and placement methods on P uptake and yield | NE | Uly SiL subsoil, Thurman LfS | O | na | Broadcast, banded | na | na | na | 4.4–11 | na | na | Placement | Sleight et al. [191] |
| Surface and subsurface P applications on corn yields and P distribution | OH | Wooster SiL | C | TSP | Broadcast, banded | 0, 14.5, 29, 19, 39, 58, 116 | PP, AP | NT | 12 | na | na | Placement | Eckert and Johnson [192] |

**Table 1.** *Cont.*

| Purpose | State † | Soil Series * | Crop ** | P Source | P Placement | P Rate (kg P ha$^{-1}$) | P Timing | Tillage | STP (mg kg$^{-1}$, kg ha$^{-1}$ *) | Tile Drain | Irrigated | Study Type | Reference |
|---|---|---|---|---|---|---|---|---|---|---|---|---|---|
| P placement depth effects on grain yield, yield components, and P uptake | NE | Holdrege Si, Hall Sit, Burchard-Shelby C, Adair-Pawnee C, Keith Si, Alliance Si, Geary SiC | WW | APP | Surface, seed placement | 11 | AP | na | 4–9 | na | na | Placement | McConnell et al. [193] |
| P placement method impact on grain yield | KS | Crete SiCL | WW | APP | Pre-plant banded, seed-banded | 0, 2.5, 5, 10, 20 | PP, AP | CT | 8–11 | na | na | Placement, rate | Cabrera et al. [194] |
| P fertilizer rate and placement effect on soybean | NE | Crofton SiL, Nora SiL, Moody SiCL | S | APP | Broadcast and incorporated, banded below/side of seed | 0, 11, 22, 33, 44 | PP, AP | na | 0.6–3.1 | na | yes | Placement, rate | Rehm [195] |
| P application methods and P sources effect on corn yields and P uptake | NE | Sharpsburg SiCL, Coly SiL | C | APP, UP, DAP | Broadcast, banded near/below seed | 9, 18 | PP | MT | 1.5–5.5 | na | yes | Source, placement, rate | Raun et al. [196] |
| P and K placement methods for NT corn | IA | Kenyon, Webster, Galva, Mahaska, Marshall, Nevin, Colo, Nicollet, Givin, Dinsdale | C | TSP | Deep banded, shallow banded, broadcast | 14, 28, 56 | PP | NT | 7–41 | na | na | Placement | Bordoli and Mallarino [23] |
| P and K fertilizer placement impact on corn growth, yield, nutrient uptake | OH | Kenyon, Webster, Galva, Mahaska, Marshall, Nevin, Colo, Nicollet, Givin, Dinsdale | C | na | Broadcast, deep banded, and shallow banded | 14, 28, 56 | PP | NT | 7–41 | na | na | Placement | Mallarino et al. [197] |
| Interaction of K fertilizer with P and N planting time fertilizer placement | SD | Lowry SiL | C | APP, 7-21-7 liquid fertilizer | Surface, with seed, close to seed furrow | 10–57 | AP | NT | 6 | na | yes | Placement, timing | Riedell et al. [198] |
| P and K fertilizer placements effect on soybean growth and nutrient uptake | IA | Kenyon, Webster, Galva, Mahaska, Marshall, Nevin, Nicollet, Givin, Dinsdale | S | na | Broadcast, banded with the planter, deep banded | 14, 28 | PP | NT | 7–39 | na | na | Placement | Borges and Mallarino [24] |

**Table 1.** *Cont.*

| Purpose | State † | Soil Series * | Crop ** | P Source | P Placement | P Rate (kg P ha⁻¹) | P Timing | Tillage | STP (mg kg⁻¹, kg ha⁻¹ *) | Tile Drain | Irrigated | Study Type | Reference |
|---|---|---|---|---|---|---|---|---|---|---|---|---|---|
| P or K fertilizers placement on soybean | IA | Dinsdale, Colo, Vesser, Downs, Webster, Clarion, Primghar | S | na | Surface, broadcast, subsurface banded at planting | 0, 19.5, 39, 78 | Annual and semi-annual | NT | na | na | Dryland | Rate, placement, timing | Buah et al. [33] |
| P and K fertilizer rates and placement | IA | Dinsdale, Vesser, Downs, Clarion, Wester, Colo | C | TSP | Broadcast, banded beside or below seed | 0, 19, 39 | AP | NT | 12–79 | na | na | Rate, placement | Buah et al. [25] |
| RT, P and K fertilizer placement effect on corn grain yield, early P and K uptake | IA | Marshall, Tama, Clarion, Canisteo, Webster | C | na | Broadcast and deep banded | 14, 56 | na | RT | 6–64 | na | na | Placement, rate, tillage | Borges and Mallarino [24] |
| Tillage, P placement and rate influence on P losses | KS | Woodson SiL | S, M | liquid fertilizer | Broadcast or knifed | 0, 24 | PP | RT, CT, NT | na | na | na | Placement, rate, runoff | (Kimmell Kimmell et al. [199] et al., 2001) |
| Management practices (manure, tillage, biosolids, inorganic fertilizer) effects on P runoff losses | WI | SiL | na | na | Surface | 71, 198, 331, 830, 441, 65 | Spring | NT, ShT, CP | na | na | na | Source, tillage, runoff | Bundy et al. [200] |
| P and K placement on soybean managed with RT | IA | Marshall, Tama, Clarion, Canisteo, Webster | S | TSP | Broadcast and deep banded | 0, 14, 56 | PP | RT | 7–61 | na | na | Placement | Borges and Mallarino [201] |
| Reduction in P runoff loss after incorporation of liquid swine manure or P fertilizer | IA | Terril loam | na | Liquid swine manure, APP | Broadcast, incorporated | 62–158 | na | CT | 24 | na | no | Source, placement, rate, runoff | Tabbara [202] |
| Crop response to VR and uniform-rate (UR) P fertilization | IA | Webster, Nicollet, Clarion | C, S | MAP | Broadcast | 35–70 | Fall | CT | 11–24 | na | na | Rate | Wittry and Mallarino [203] |
| Interaction effects of deeper P and K fertilizer placement with hybrid and planting population | IN | Toronto-Millbrook complex, Drummer soils | C | DAP | Broadcast, deep banded, shallow banded | 44 | PP | CT | na | na | Dryland | Placement | Kline [30] |
| P and K starter fertilizer placement effects on corn yield and nutrient uptake | IA | Sparta, Marshall, Readlyn, Marshan, Webster, Atterberry | C | 3–8–15 (N–P–K) liquid, TSP | Broadcast and in-furrow | 5–7; 49–66 | Starter | NT, CP | 5–77 | na | na | Placement | Kaiser et al. [204] |

**Table 1.** *Cont.*

| Purpose | State † | Soil Series * | Crop ** | P Source | P Placement | P Rate (kg P ha$^{-1}$) | P Timing | Tillage | STP (mg kg$^{-1}$, kg ha$^{-1}$ *) | Tile Drain | Irrigated | Study Type | Reference |
|---|---|---|---|---|---|---|---|---|---|---|---|---|---|
| STP trends over time for different initial STP levels and response of corn and soybean yield to P fertilization and STP | IA | Nicollet-webster complex, Webster–Canisteo complex, Kenyon | C, S | TSP | Broadcast | 0, 22, 33, 44 | Fall | CT | 17–96 | na | na | Rate | Dodd and Mallarino [205] |
| Tillage and annual P fertilizer management on stratified soils on plant growth and P uptake. | KS | Parsons SiL, Catoosa SiL | C, S, W | APP | Broadcast, deep banded | 0, 20 | PP | MP, NT, ReT | 16–27 | na | na | Rate, placement, tillage | Schwab et al. [26] |
| Spatio-temporal variations of corn-soybean yield and economics of variable rate N and P management | MN | Jeffers CL series, Clarion-Swanlake CL, Webster-Delft CL | C, S | TSP | na | 25, 49 | Fall | na | <5 to >15 | na | na | Rate | Lambert et al. [206] |
| P application rates (fixed rate vs. variable rate) influence on corn and soybean yield | IA | Clarion, Webster, Canisteo, Marshall | C, S | MAP | Broadcast | 24–70 | Fall | CT | 8–27 | ns | na | Rate | Bermudez and Mallarino [207] |
| P and/or K placement effect on corn growth, development and yield | IN | Drummer, Raub-Brenton complex | C | TSP, MAP, APP | Broadcast, deep banded | 44 | Fall, PP | ST | 23–109 | yes | na | Placement | Cánepa [208] |
| P availability from manure to crop growth through crop P uptake and yield; residual P availability from manure application during consequent year; evaluate the effect of P source on changes in STP levels. | WI | Plano SiL, Withee SiL | C | Various manure, TSP | | 39, 79, 118 | PP | CT | 11–12 | na | na | Source | Sneller and Laboski [209] |
| Impact of fall and spring broadcast P fertilization on P uptake and grain yield | IA | Multiple | C, S | TSP | Broadcast | 0, 10, 20, 30, 40, 50 | Fall, spring | NT | 5.1–34 | na | na | Timing | Mallarino et al. [35] |
| In-furrow fluid starter P–K fertilizer application impact on yield, P and K concentration and uptake | IA | Sparta LS, Readlyn L, Marshan CL, Webster SiCL | C | 3–8–15 fluid fertilizer, TSP | Broadcast, starter | 5–7; 49–66 | Starter | CT, NT | 4–56 | na | na | Placement, source | Mallarino et al. [210] |

**Table 1.** *Cont.*

| Purpose | State † | Soil Series * | Crop ** | P Source | P Placement | P Rate (kg P ha⁻¹) | P Timing | Tillage | STP (mg kg⁻¹, kg ha⁻¹ *) | Tile Drain | Irrigated | Study Type | Reference |
|---|---|---|---|---|---|---|---|---|---|---|---|---|---|
| Tillage, P and K fertilizer rate and placement effect on soybean roots distribution, soil water, P, and K levels. | IL | Drummer SiCL, Flanagan SiL | S | na | Broadcast, deep banded | 0, 12, 24, 36 | PP | NT, ST | 20 | yes | na | Tillage, placement and rate | Farmaha et al. [32] |
| Effect of P and K rate and placement in NT and ST on P and K accumulation | IL | Drummer SiCL, Flanagan SiL | S | na | Broadcast, deep banded | 0, 12, 24, 36 | na | NT, ST | 21 | yes | na | Rate, placement | Farmaha et al. [31] |
| P and K distribution after repeated applications in NT and ST soils | IL | Drummer SiCL, Flanagan SiL | | DAP, TSP | Broadcast, deep banded | 22, 333, 44, 55, 66, 77 | | NT, ST | na | na | na | Tillage, rate and placement, stratification | Fernández and Schaefer [28] |
| Effect of P and K rate and placement in NT and ST on grain yield; soil water, P, and K content, corn roots distribution | IL | Drummer SiCL, Flanagan SiL | C | TSP | Broadcast, deep banded | 0, 12, 24, 36 | PP | NT, ST | 41 | na | na | Rate, placement, tillage | Fernández and White [29] |
| Effect of starter and broadcast fertilizer application on corn and soybean production, STP | KS | Eudora SL, Rossville SL, Woodson SL; Kenoma SL; Crete SL | C, S | MAP | Starter, broadcast | 9.8, 19.6, 29.3, 39.1, 48.9 | Starter, PP | CT, NT | 12–26 | na | Irrigated, rainfed | Placement, timing | Arns [211] |
| Review of tillage system and P fertilizer placement interaction on corn and soybean production. | KS | Woodson SiL, Crete SiL | C, S | TSP, APP | Broadcast, deep banded | 0, 20, 39 | Starter, AP | CT, NT | na | na | Irrigated, rainfed | Tillage, P placement | Edwards [212] |
| Tillage, P placement and rate impact on P runoff | IL | Drummer SiCL, Flanagan SiL | C, S | TSP | Broadcast, deep banded | 23, 40 | | NT, ST | 12–31 | no | | Tillage, placement, rate | Yuan et al. [61] |

\* SiL, silt loam; SiC, silty clay; SiCL, silty clay loam; SL, sandy loam; Si, Silt; LS, loamy sand; L, loam; CL, clay loam; Si, silt; LfS, loamy fine sand; fSL, fine sandy loam. \*\* C, corn; S, soybean; M, sorghum; O, oat; WW, winter wheat; SW, spring wheat; H, hay; A, alfalfa; CC, cover crops. Abbreviations: APP, ammonium polyphosphate; CP, chisel plow; CT, conventional tillage; DAP, diammonium phosphate; DT, disk tillage; MAP, Monoammonium phosphate; MP, Moldboard plow; MT, minimum tillage; na, not available; NT, no-tillage; OA, orthophosphoric acid; P, phosphorus; RP, rock phosphate; Ret, reduced tillage; RT, ridge tillage; ShT, shallow tillage; STP, soil test phosphorus; SP, superphosphate; ST, strip tillage; TSP, triple superphosphate; UP, urea phosphate. † States: OH, Ohio; IL, Illinois; IA, Iowa; MI, Michigan; MO, Missouri; MN, Minnesota; NC, North Carolina; IN, Indiana; KS, Kansas; SD, South Dakota; NE, Nebraska; ND, North Dakota; KY, Kentucky; VA, Virginia.

**Table 2.** Studies evaluating P stratification over time.

| Purpose | State | Soil Series | Crop Rotation | P rate (kg ha$^{-1}$) | Tillage | STP (mg kg$^{-1}$) | Reference |
|---|---|---|---|---|---|---|---|
| Comparison of P stratification between CT, NT, RiT | IN | Chalmers SiCL | C | 274 (biannual) | CT, NT, RiT | 35 to 117 | Mackay et al. [213] |
| Effects of 8-yr P buildup and 26-yr residual decline on crop yields and soil-test P. | NC | Portsmouth soil | C-S | 0, 10, 20, 40, and 60 (annually) | CT | Initial STP 22 g m$^{-3}$; adding 30 g P m$^{-3}$ yr$^{-1}$ resulted in an increase of 7.4 g P m$^{-3}$ yr$^{-1}$ | McCollum [36] |
| Effects of P and K fertilization on STP | IA | Kenyon L | C-S | 0, 22, 45 | na | 28 | Mallarino et al. [214] |
| Vertical and horizontal distributions of P in conservation tillage systems | IA | Webster CL, Tama SiL | C-S, CCo | 30, 80 | NT, RiT | 69 to 129 | Robbins and Voss [38] |
| Changes in soil chemical properties, associated with different crop rotation and tillage practices over a 12-yr period | IA | Floyd L, Kenyon L, Readlyn L | C-S, CCo | 17 to 58 | MP, CP, RiT, NT | 57 to 141 | Karlen et al. [215] |
| changes in STP values, crop yields and economic returns to P fertilization resulting from 14-yr of annual applications of P fertilizer | IA | Webster CL, Canisteo CL | C-S | 0, 11, 22, 34 | na | 18 | Webb et al. [216] |
| P build up and decline was determined during a 20-yr period and critical STP concentrations were determined for corn and soybean | MN | Webster CL, Aastad CL | C-S | 0, 56, 112 | na | Initial STP 10 mg kg$^{-1}$; adding 56 kg P ha$^{-1}$ resulted in an increase of 0.7 mg P kg$^{-1}$ yr$^{-1}$ and adding 112 kg P ha$^{-1}$ resulted in an increase of 2.5 mg P kg$^{-1}$ yr$^{-1}$ | Randall et al. [37] |

**Table 2.** *Cont.*

| Purpose | State | Soil Series | Crop Rotation | P rate (kg ha$^{-1}$) | Tillage | STP (mg kg$^{-1}$) | Reference |
|---|---|---|---|---|---|---|---|
| **Long-term tillage management impact on P fractions in the soil** | MI | Capac L, Kalamazoo L | C-S, CCo | na | CT, NT | 32 to 107 | Daroub et al. [217] |
| Determining changes in soil P dynamics over time in Sanborn field. | MO | Mexico SiL | CCo, CW, CT, C-W-RC | 0–31 | MP | 0–75 | Motavalli and Miles [19] |
| **P stratification after deep banding fertilizers for 4 yr** | IA | Kenyon, Webster, Galva, Mahaska, Marshal | C-S | 28, 66 | NT, CT | 12 to 56 | Mallarino and Borges [39] |
| **Survey of P, K, pH, Ca, Mg, and organic matter levels of soils in Illinois and the degree of nutrient vertical stratification.** | IL | na | na | na | na | 1 to 576 | Fernández et al. [218] |
| Effects of 45 yr of fertilizer and tillage treatments on soil nutrients and crop yields | IL | Bethalto SiL | C-S, C | 14 to 39 | MP, CP, NT | 5 to 35 | Cook and Trlica [18] |
| **P stratification at the watershed scale and its relationship to STP, and potential contribution to increased DRP export** | OH | na | C, S, W | na | 65% NT | na | Baker et al. [43] |

Abbreviations: CP, chisel plow; CT, conventional tillage; MP, moldboard plow; na; not available; NT, no-tillage; P, phosphorus; RiT, ridge tillage; STP, soil test phosphorus; C-S, corn-soybean rotation; CCo, continuous corn rotation; CW, continuous wheat; CT, continuous timothy; C-W-RC, corn-wheat-red clover. † States: OH, Ohio; IL, Illinois; IA, Iowa; MI, Michigan; MO, Missouri; MN, Minnesota; NC, North Carolina; IN, Indiana.

**Table 3.** Fertilizer amendments used for managing phosphorus.

| Purpose | State † | Soil Series | Amendment or Enhance P Fertilizer Type Rate | | Crop | P rate | STP (mg kg$^{-1}$, kg ha$^{-1}$ *) | Study type | Highlights | Reference |
|---|---|---|---|---|---|---|---|---|---|---|
| **Effectiveness of soil amendments on reduction of drainage water P concentration.** | FL | Pahokee muck | Calcium oxide plus aluminum sulfate, dolomite, gypsum | 0, 4, 8, and 12 Mg ha$^{-1}$ | na | 5 mg L$^{-1}$ | 5 | Column leaching | 25–40% reduction of DP with gypsum compared to other treatments | Coale et al. [44] |
| **Coal combustion by-products and gypsum effects on heavy metal uptake and P loss in surface runoff** | PA | Klinesville, Hagerstown, Watson | Flyash, FGD gypsum, agriculture gypsum | 5, 10, and 20 g kg$^{-1}$ | Canola | na | 128 to 370 | Growth chambers, runoff boxes | 20–43% reduction in DP | Stout et al. [219] |
| **Effect on P sorption capacity of Ap horizon after applying limestone, dolomite, and gypsum** | FL | Immokalee fS | Limestone, dolomite, gypsum | 1.8 Mg ha$^{-1}$ for gypsum and 1 Mg ha$^{-1}$ for other two | Pasture | na | 2 | Column study | Ca amendments that increase soil pH are more efficient at retention of P in soil | Boruvka and Rechcigl [220] |
| **Repeated plant growth cycles impact on the stability of soil inorganic P fractions formed after FGD gypsum application** | DE, PA | Watson SiL, Klinesville SiL | FGD gypsum | 22 Mg ha$^{-1}$ | Ryegrass | na | 228 to 367 | Greenhouse | Treatment with FGD decreased water extractable soil P 38% to 57%, | Stout et al. [221] |
| **Alum amended poultry manure effects on P release from soils** | DE | Evesboro LS, Rumford LS, Pocomoke SL | Aluminum sulfate amended poultry manure | 9 Mg ha$^{-1}$ | na | na | 467 to 671 | Controlled incubation | 7.3% to 20% reduction in P desorption from amended soils compared to control | Staats et al. [45] |

**Table 3.** *Cont.*

| Purpose | State † | Soil Series | Amendment or Enhance P Fertilizer Type Rate | | Crop | P rate | STP (mg kg$^{-1}$, kg ha$^{-1}$ *) | Study type | Highlights | Reference |
|---|---|---|---|---|---|---|---|---|---|---|
| **P removal efficiencies of two amendments with or without limestone** | VT | na | Electric arc, furnace steel slag, serpentinite | na | na | na | na | Column study | Serpentinite + limestone removed 1.0 mg P g$^{-1}$ and steel slag + limestone removed 2.2 mg P g$^{-1}$ of material used during 180 d of experiment | Drizo et al. [222] |
| **effectiveness of grass buffer strips and gypsum amendments in reducing the P loss from land-applied poultry litter** | AL | Hartsells fSL | Gypsum | 0, 1, 3.2, and 5.6 Mg ha$^{-1}$ | Tall fescue | 11 kg ha$^{-1}$ | na | Grass buffer strip, simulated runoff | 32–40% reduction in DP in grass buffer strips with gypsum | Watts and Torbert [51] |
| **Increasing levels of gypsum application effects on solubility of 13 nutrients** | NE | Sharpsburg SiCL | Gypsum | 0, 0.01, 0.05, 0.1, 0.15, 0.2, 0.3, and 0.5 g g$^{-1}$ of soil | na | na | na | Laboratory experiment | Gypsum addition increased the solubility of N, K, Ca, Mg, Mn, Cl, and S, whereas it decreased the solubility of P, Na, Fe, Cu, Zn, and B. | Elrashidi et al. [223] |
| **Effect of liming, P source, and P enhancer products on corn production and P uptake** | MO | Putnam SiL, Tiptonville SiL | Limestone, Avail, P$_2$O$_5$ Max | Limestone (3.4, 4.5, and 8.1 Mg ha$^{-1}$) and Avail and P$_2$O$_5$ Max (2.1 and 4.2 L Mg$^{-1}$ of fertilizer) | Corn | 24, 49. and 51 kg ha$^{-1}$ | 30 to 118 * | Field | P enhancers did not affect plant population, silage dry weights, grain moisture, yield, protein, oil, or starch | Dudenhoeffer et al. [55] |

**Table 3.** *Cont.*

| Purpose | State † | Soil Series | Amendment or Enhance P Fertilizer Type Rate | | Crop | P rate | STP (mg kg⁻¹, kg ha⁻¹ *) | Study type | Highlights | Reference |
|---|---|---|---|---|---|---|---|---|---|---|
| Effectiveness of an in-ditch filter to remove DP was evaluated | MD | Quindocqua SiL, Manokin SiL | FGD gypsum | 110 Mg | na | na | 374 * | Filter in drainage ditch | 65% to 73% DP removal. Ditch filtration using FGD gypsum is not practical at a farm scale due to maintenance and clean-out requirements | Bryant et al. [224] |
| Leaching potential of P after application of gypsum amendments and different levels of exchangeable Ca2+ and Mg2+ to the soil | IN | Miami SiL | Gypsum, 5 Ca:Mg ratios | 5 Mg ha⁻¹ | na | 45 kg ha⁻¹ | 53 | Column study | Leaching of particulate P was significantly less in the Ca-treated soil than the Mg-treated soil | Favaretto et al. [225] |
| Effect of tillage, fertilizer placement, P rate, and two P enhancer products on corn production, grain quality, P uptake, and apparent P recovery efficiency | MO | Kilwinning SiL, Bremer SiCL | Avail, $P_2O_5$ Max | 2.1 and 4.2 L Mg⁻¹ of fertilizer | Corn | 0, 24, and 49 kg ha⁻¹ | 27 to 90 * | Field | P enhancers addition did not increase plant P uptake | Dudenhoeffer et al. [47] |
| Reduction in P losses with application of FGD gypsum | AL | Luverne SL | FGD gypsum | 0, 2.2, 4.4, and 8.9 Mg ha−1 | Bermudagrass | 20.6 g P kg⁻¹ (13.4 Mg ha⁻¹ poultry litter wet wt.0 | na | Simulated runoff | 54% cumulative reduction in DP concentration losses was observed with FGD gypsum compared control | Watts and Torbert [53] |

Table 3. *Cont.*

| Purpose | State † | Soil Series | Amendment or Enhance P Fertilizer Type Rate | | Crop | P rate | STP (mg kg$^{-1}$, kg ha$^{-1}$ *) | Study type | Highlights | Reference |
|---|---|---|---|---|---|---|---|---|---|---|
| **Impact of FGD gypsum on P concentrations and loads in surface runoff and tile discharge** | OH | Blount SiL | FGD gypsum | 2.24 Mg ha$^{-1}$ | Continuous corn | 0 kg ha$^{-1}$ | >480 | Field runoff, tile drainage | Combined surface and tile discharge reduction of DRP and TP were 36% and 38%. FGD gypsum can be used as a tool to address elevated P concentrations and loadings in drainage waters. | King et al. [52] |
| **Gypsum effects on crop yield, STP, plant tissue P, and vadose water** | AL, AR, IN, NM, ND, OH, WI | na | FGD, mined gypsum | | Alfalfa, Bermudagrass, canola, cotton, corn, soybean, wheat | 0 to 22.4 Mg ha$^{-1}$ | na | Field | Crop yield was generally not affected by gypsum application however reduction in DP losses in water were seen | Kost et al. [49] |
| **impacts on soil, plant tissue, and surface water runoff from fields receiving FGD gypsum** | AL | Luverne sandy | FGD gypsum | 0, 2.2, 4.4, and 8.9 Mg ha$^{-1}$ | Bermudagrass | na | 30 | Simulated runoff | FGD gypsum application did not result in increase of toxic elements in plants, soil, or runoff | Torbert et al. [226] |

Abbreviations: DP, dissolved P; FGD, flue gas desulfurization; na, not available; P, phosphorus; STP, soil test phosphorus. †States: OH, Ohio; IL, Illinois; IA, Iowa; MI, Michigan; MO, Missouri; MN, Minnesota; NC, North Carolina; IN, Indiana; KS, Kansas; SD, South Dakota; NE, Nebraska; ND, North Dakota; KY, Kentucky; VA, Virginia; FL, Florida; PA, Pennsylvania; DE, Delaware; VT, Vermont; AL, Alabama; MD, Maryland; AR, Arkansas.

**Table 4.** Cover crops and reduced tillage as best management practices for managing phosphorus loss.

| Purpose | State † | Soil Series | Crop | Tillage | STP (mg kg$^{-1}$, kg ha$^{-1}$ *) | P rate (kg ha$^{-1}$) | P source | P placement | P timing | Study Type | Highlights | Reference |
|---|---|---|---|---|---|---|---|---|---|---|---|---|
| **Effects of tillage types on corn yield, P accumulation and soil compaction** | VA | Lodi loam | C, Annual ryegrass, O, rye | CT, NT | na | 22–49 | 5-10-5, 10-100-10 15-10-10 | Broadcast, incorporated | | Tillage, cover crops | At same application rate of P, available P accumulation was more in the upper 5 cm of the untilled soil than NT. NT soil had greater available phosphorus for the upper 20 cm of soil | Shear and Moschler [179] |
| **Crop rotation, soil management practices and fertilizer rates impact on soluble N and P losses in surface runoff.** | NY | Lima-Kendaia soil association | C, beans, wheat, rye, A | na | | 10–49 | na | Broadcast, planter, sidedressed | | Runoff, cover crops | Three times higher P losses in high fertility, poorly managed plots than other treatments. | Klausner et al. [227] |
| **Tillage effects on runoff quality and quantity** | MD | Manor loam | C, barley | CT, NT | 87 | 108 | 6-24-24 | na | | Runoff, cover crops | Higher runoff, sediment and soluble solids losses from CT than NT. 163 g ha$^{-1}$ more TP lost from CT than NT in one year. No significant differences between CT and NT for loss of ortho-PO, and total soluble P | Angle et al. [228] |

**Table 4.** *Cont.*

| Purpose | State † | Soil Series | Crop | Tillage | STP (mg kg$^{-1}$, kg ha$^{-1}$ *) | P rate (kg ha$^{-1}$) | P source | P placement | P timing | Study Type | Highlights | Reference |
|---|---|---|---|---|---|---|---|---|---|---|---|---|
| **Conservation practices impact on runoff P forms** | GA | Cecil SL | C, S, M WW, barley, crimson clover, rye | CT, RT | | 20–50 | | Preplant-incorporated, broadcast | | Runoff, cover crops | higher soluble P and total P concentrations and >50% lower TP losses with conservation tillage than CT. Lower runoff losses with conservation tillage RT with CC most effective in decreasing runoff, sediment, and nutrient losses. | Langdale et al. [229] |
| **Tillage and cover crop impacts on runoff** | AL | Decatur SiL | Cotton, WW | NT, RT, CT | na | na | na | na | | Runoff, cover crops | Chickweed, downy brome, and Canada bluegrass decreased annual soil losses by 87%, 95%, and 96%, and runoff by 44%, 53%, and 45%, respectively, compared to no-CC control. CC have 1.62 to 2.86-time greater dissolver phosphate than no-CC control. CC reduced annual dissolved nutrient losses by 7% to 77% | (Yoo et al. [230] |
| **Effectivenessof selected non-leguminous winter cover crops in reducing runoff, soil loss, and dissolved N and P levels transportedin runoff.** | MO | Mexico SiL | S, common chickweed, Canada bluegrass, downy brome | NT | na | 25 | 6-10-20 | na | | Runoff, cover crops | | Zhu et al. [78] |

**Table 4.** *Cont.*

| Purpose | State † | Soil Series | Crop | Tillage | STP (mg kg$^{-1}$, kg ha$^{-1}$ *) | P rate (kg ha$^{-1}$) | P source | P placement | P timing | Study Type | Highlights | Reference |
|---|---|---|---|---|---|---|---|---|---|---|---|---|
| **Effect of rye CC and N fertilizer sources on soil chemical properties** | OH | Canfield SiL, Hoytville SiC | C, S, rye, A | NT | 0.20–0.54 mmol/kg P | 26–38 | na | Broadcast | Spring | Cover crops | Rye CC reduced P concentration in surface 5 cm soil depth in continuous corn rotation that received ammonia fertilizer applications. | Eckert [71] |
| **Effects of winter CC on soil chemical and physical properties** | IL | Flanagan | C, S, hairy vetch, cereal rye | NT | | Not applied | no | no | no | Cover crops | Lower soil P in rotation with rye or mixture of rye and hairy vetch CC compared to no-CC treatments. | Villamil et al. [73] |
| **CCs impact on P uptake and soil P concentration** | MD | Downer, Codorus | C, forage radish, cereal rye | CT | 88–98 | 17 | TSP | Broadcast | In-season | Cover crops | Forage radish resulted into 19 and 22 mg P kg$^{-1}$ more P in 0–2.5 cm soil depth than cereal rye and No-CC control | White and Weil [231] |
| **Compaction and CC effects on soybean growth, yield, and soil properties** | IL | Drummer SiCL | S, radish, triticale, buckwheat, hairy vetch | CT | na | no | Not used | no | no | Cover crops | No differences obtained in soil P concentration due to CC | Acuña and Villamil [232] |
| **Effects of single or mixture of CCs on crop yields, weeds and soil properties** | IL | Flanagan SiL, Danabrook SiL | C, S, forage radish, buckwheat, cereal rye, hairy vetch | CT | | 1 ton | Manure (5-3-3) | | na | Cover crops | In non-headland areas, mixture of forage radish, hairy vetch and rye had lower soil P by 6.4 mg/kg than in forage radish + buck wheat treatment. | Welch et al. [177] |

**Table 4.** *Cont.*

| Purpose | State † | Soil Series | Crop | Tillage | STP (mg kg$^{-1}$, kg ha$^{-1}$ *) | P rate (kg ha$^{-1}$) | P source | P placement | P timing | Study Type | Highlights | Reference |
|---|---|---|---|---|---|---|---|---|---|---|---|---|
| **Impact of tillage and CCS on crop yield and soil properties** | IL | Flanagan SiL, Drummer SiCL, Catlin SiL | C, S, rape, radish, annual ryegrass, red clover, hairy vetch, cereal rye, spring oat | NT, CP | | na | na | na | na | Cover crops | Available P not affected by CC or tillage | Dozier et al. [233] |
| **Performance of winter CCs in reducing nitrate and TP in tile-drained agricultural watershed. effectiveness of winter cover crops in** | IL | na | C, S, cereal rye, tillage radish, O, annual rye | | | na | na | na | na | Watershed scale, cover crops, subsurface tiles | No significant change in TP loading | Bruening [77] |
| **reducing runoff, total suspended solids (TSS), nitrogen (N) and phosphorus (P) concentrations in ephemeral streams of non-tile drained headwater agricultural watersheds.** | IL | Hosmer SiL | C, S, cereal rye, hairy vetch | NT | na | 26 | DAP | Broadcast | na | Cover crops | Event meant concentrations of NO$_3$-N, NH$_4$-N, and DRP did not decrease | Singh et al. [67] |
| **impacts of cover crops on the water quality draining from cotton production fields.** | AR | Mhoon fine SL, Commerce vfSL, Bruno LS, Dundee fSL, Routon, Crevasse, Sharkey-Steele complex | Cotton, C, O, WW | CT | na | 22–28 | na | Broadcast | na | Cover crops, runoff | CC reduced phosphate by 53% than control at one of the sites | Aryal et al. [234] |

**Table 4.** *Cont.*

| Purpose | State † | Soil Series | Crop | Tillage | STP (mg kg$^{-1}$, kg ha$^{-1}$ *) | P rate (kg ha$^{-1}$) | P source | P placement | P timing | Study Type | Highlights | Reference |
|---|---|---|---|---|---|---|---|---|---|---|---|---|
| effects of cover crop in a corn-soybean rotation on nutrient loss, soil health, and crop yields in a terraced field | MO | Putman SiL | C, S, WW, radish, turnip and cereal rye | NT | 80–90 | 28 | na | Broadcast | Spring | Cover crops and terraces | CC did not decrease cumulative total P loss | Adler [98] |
| Effects of different N application rates and tillage practices on corn yield | OH | Canfield SiL | C | NT, CT | 30* | 9 +55 | TSP | Broadcast, mixed | At planting | Tillage | NT increased corn yields compared to CT, but no significant differences in P uptake at tasseling stage | Triplett and Van Doren [180] |
| Tillage practices impact on corn growth, and P stratification | VA | Lodi loam | C | NT, CT, tillage alternate year | na | 112 | 10/10/2010 | Broadcast and/or incorporated | na | Tillage, cover crops | NT increased corn yields than CT. | Shear and Moschler [179] |
| tillage methods effects on runoff water and sediment N and P composition | IN | Bedford SiL | C | Coulter plant, till-plant, chisel-plant, DT, CT | na | 56 | TSP | na | na | Tillage | CT had highest soil erosion and water losses, but small N and P losses; disk and till systems have lower soluble N and P concentrations in runoff water. | Römkens and Nelson [62] |
| Tillage and P placement methods impact on corn growth and yield. | IL, NE | Sharpsburg SiCL, Leshara SiL, Platte SL, Flanagan SiL | C | CP, MP | 4.2–12.8 | 0, 11, 22, 45, 30, 60 | OA | Broadcasting, banding, application by CP | Pre-planting | Placement, rate, tillage | Lower P losses in runoff with chisel placement of P | Cihacek et al. [185] |

**Table 4.** *Cont.*

| Purpose | State † | Soil Series | Crop | Tillage | STP (mg kg$^{-1}$, kg ha$^{-1}$ *) | P rate (kg ha$^{-1}$) | P source | P placement | P timing | Study Type | Highlights | Reference |
|---|---|---|---|---|---|---|---|---|---|---|---|---|
| **P fertilizer application and tillage system impacts on corn production and P movement in soil profile** | IL | Clinton and Ipava SiL | C | CT, NT | na | 51.5 | na | Broadcast | Pre-planting | Tillage | Slow P movement in soil under NT than CT. | Fink and Wesley [235] |
| **N and P runoff and sediment losses as affected by tillage practices** | IA | Ida SiL, Tama SiCL, Kenyon SL | C | CT, till plant, CP, DT, ridge-plant, fluted coulter | na | 30 | na | Broadcast | Before spring tillage and planting | Runoff, tillage | Conservation tillage practices reduced soil erosion and related nutrient losses, but they were not effective in reducing the loss of water soluble nutrients | Barisas et al. [236] |
| **Comparison of conservation tillage system with CT for sediment and nutrient losses in runoff** | IA | Loess, Hills, Monona-Ida-Napier soils | na | Conventional plowing and planting, till-planting, and ridge-planting | 22 | 37 | na | na | na | Tillage, runoff, placement | Conservation tillage reduced runoff (40%), soil loss (60–90%), N and P losses as compared to Conventional plowing. | Johnson et al. [79] |
| **Tillage practices effects on N and P losses in runoff in corn-soybean rotation** | IA | Clarion, Monoma | C, S | CT, CP, NT | na | 37 | AP | Broadcast | Before tillage and planting | Tillage and runoff | Phosphate-P in runoff water and sediment TP follows the order: NT > CP > CT. P loses in erosion followed opposite trend. | Laflen [237] |

**Table 4.** *Cont.*

| Purpose | State † | Soil Series | Crop | Tillage | STP (mg kg$^{-1}$, kg ha$^{-1}$ *) | P rate (kg ha$^{-1}$) | P source | P placement | P timing | Study Type | Highlights | Reference |
|---|---|---|---|---|---|---|---|---|---|---|---|---|
| **Effects of tillage systems on runoff P loss were evaluated** | WI | Griswold SiL | C | CT, TP, NT, CP | 39–58 | 111 | 6–10.6–20; 6-24-24 | Subsurface banded | At planting | Tillage, runoff | All conservation tillage systems reduced TP and AAP losses by 59% to 81% and 27% to 63%, respectively, than CT. | Andraski et al. [238] |
| **Impact of NT on P-retention in soil** | OH | Hoytville SiCL, Canfield SiL | na | NT, CT | na | na | na | na | na | Tillage | na | Guertal et al. [239] |
| **influence of crop rotation under different tillage practices on soil erosion, N and P export using the EPIC model** | IL | na | soybean, S | CT, NT | na | na | na | na | na | Tillage, crop rotation | NT resulted higher P losses in surface runoff | Phillips et al. [240] |
| **Conservation tillage impact on soil erosion, N and P losses in runoff.** | KY | Maury SiL | na | CT, CP, NT | na | 44 | TSP | Broadcast | na | Tillage, runoff | NT had lower mean runoff rate, total runoff volume, mean sediment concentration, and total soil losses compared to CP and CT. NT increased phosphate concentration in runoff than CT or CP. | Seta et al. [241] |
| **Interaction effect of tillage systems and crop rotation on P stratification** | IN | Chalmers SiCL | C, S | MP, CP, NT | na | na | na | na | na | Tillage, crop rotation, P stratification | Characteristic P stratification in NT due to surface fertilizer application | Holanda et al. [242] |

**Table 4.** *Cont.*

| Purpose | State † | Soil Series | Crop | Tillage | STP (mg kg⁻¹, kg ha⁻¹ *) | P rate (kg ha⁻¹) | P source | P placement | P timing | Study Type | Highlights | Reference |
|---|---|---|---|---|---|---|---|---|---|---|---|---|
| **Manure and compost application impact on runoff losses of P and N** | NE | Sharpsburg SiCL | M, WWt | NT, DT | 12–79 | na | Manure | na | na | Tillage, placement, runoff | NT had lower TP and PP concentrations than disked treatments. DP, BAP losses in runoff were greater with NT than Disked treatments. Under very low STP levels, large responses to P were observed for all placements. | Eghball and Gilley [243] |
| **Corn response to P placement and rates under various tillage practices** | MN | Nicollet-Webster CL | C | na | na | na | na | na | na | 4R, tillage | Banded applications at half the recommended broadcast rate was not enough to optimize corn grain yield | Randall et al. [244] |
| **Effects of tillage and N/P source on surface runoff losses of N and P fractions.** | MN | Webster CL | C | MP, RT | na | 53–86 | TSP, manure | Surface broadcast | Fall, spring | Tillage, runoff, subsurface tile | RT with manure applications increased TP and soluble P losses. MP with manures resulted into least water quality degradation | Zhao et al. [245] |

Table 4. *Cont.*

| Purpose | State † | Soil Series | Crop | Tillage | STP (mg kg⁻¹, kg ha⁻¹ *) | P rate (kg ha⁻¹) | P source | P placement | P timing | Study Type | Highlights | Reference |
|---|---|---|---|---|---|---|---|---|---|---|---|---|
| **Impact of manure application and tillage on runoff P losses** | WI | Plano and Rozetta SiL | C | CP, NT | na | 88 | manure | Broadcast | Spring | Tillage, runoff | NT resulted into greater P stratification near the surface (0–5 cm) than CP. NT reduced P loads by 57%, 70% and 91% for dissolved P, bioavailable P, and TP, respectively as compared to CP. | Andraski et al. [246] |
| **Impact of tillage and starter fertilizer on grain yield and nutrient uptake** | IA | Maxfield, Donnan, Marshall, Klinger, Sawmill, Dinsdale | C | ST, DT | 14–50 | 5.2–24.2 | 6-8-6, 7-8-5, 10-15-0, 16-10-3 | Broadcast | na | Tillage | Tillage increased yield and nutrient uptake by 2.5% and 22–30%, respectively. | Bermudez and Mallarino [247] |
| **Effects of P fertilizer management under different tillage systems on crop yield and P uptake** | KS | Parsons SiL, Catoosa SiL | C, S, WW, M | ReT, NT, MP | 12–27 | 20 | APP | Broadcast, banding, deep banding | Pre-planting | 4R, tillage | Corn and sorghum yield and P uptake were increased with subsurface placement of P. MP increased grain yields of corn, soybean and wheat as compared to NT | Schwab et al. [26] |
| **Evaluate interactive effects of tillage systems and N rates of liquid swine manure and N fertilizer on corn N and P use efficiencies** | IA | Kenyon loam | C | NT, ST, CP | 35 | na | Liquid swine manure | na | na | Tillage | Greater P recovery with CP than NT or ST with manure application at 85 kg N ha⁻¹. For N fertilizer treatments, NT had greater grain P recovery than ST or CP at all N rates. | Al-Kaisi and Kwaw-Mensah [248] |

**Table 4.** *Cont.*

| Purpose | State † | Soil Series | Crop | Tillage | STP (mg kg$^{-1}$, kg ha$^{-1}$ *) | P rate (kg ha$^{-1}$) | P source | P placement | P timing | Study Type | Highlights | Reference |
|---|---|---|---|---|---|---|---|---|---|---|---|---|
| **Influence of tillage and fertilizer N-P management on short-season corn grown** | KS | Parsons SiL | C | ST, NT, ReT | 17 | 20 | APP | Surface band, subsurface band | Spring, fall | Tillage | ReT increased corn yields by 2.82 Mg ha$^{-1}$ than other tillage systems. Spring and subsurface banding applications increased yields than other treatments. | Sweeney et al. [249] |
| **Effect of P and K rate and placement in NT and ST on grain yield; water, P, and K values in the soil; and the distribution of corn roots were evaluated.** | IL | Flanagan SiL, Drummer SiCL | C | NT, ST | 41 | 0,12,24,36 | TSP | Broadcast, deep banding | Pre-planting | Tillage, rate, placement | Deep banding increased soil P beneath the crop row and reduced soil surface test values compared to broadcast applications | Fernández and White [29] |
| **Effect of crop rotation and tillage on both soil chemical and physical properties** | IL | Sable silty CL, Muscatune SiL, Caseyville SiL, Downsouth SiL | C, S, WW | CP, NT | | na | na | na | na | Tillage, crop rotation | NT had 8.8 mg kg more P in soil than CT at depth 0–10 cm. However, at depth 10–20 cm, NT had 3.2 mg kg$^{-1}$ less P than CT. continuous soybean rotation had higher soil P concentrations than continuous corn a corn-soybean-wheat | Zuber [250] |

**Table 4.** *Cont.*

| Purpose | State † | Soil Series | Crop | Tillage | STP (mg kg⁻¹, kg ha⁻¹ *) | P rate (kg ha⁻¹) | P source | P placement | P timing | Study Type | Highlights | Reference |
|---------|---------|-------------|------|---------|---------|---------|----------|-------------|----------|------------|------------|-----------|
| **Effects of tillage, P fertilizer placement and rate on runoff P concentrations and loads** | IL | Drummer SiCL, Flanagan SiL | C, S | NT, ST | 12–31 | 23, 40 | TSP | broadcast, deep placement | Fall | 4R, runoff, tillage | DRP loads reduced by deep placement 69% to 72% compared to broadcast P application, irrespective of rate. Increasing P rates increased P concentration for broadcast treatments. Deep placement also reduced TP runoff losses | Yuan et al. [61] |

Abbreviations: AP, ammonium phosphate; APP, ammonium polyphosphate; BAP, bioavailable phosphorus; CC, cover crop; CP, chisel plow; CT, conventional tillage; DRP, dissolved reactive phosphorus; MP, moldboard plow; na, not available; NT, no-tillage; OA, orthophosphoric acid; P, phorphorus; PP, particulate phosphorus; ReT, reduced tillage; RT, ridge tillage; ST, strip tillage, STP, soil test phosphorus; TSP, triple super phosphate; TP, total phosphorus. † States: OH, Ohio; IL, Illinois; IA, Iowa; MI, Michigan; MO, Missouri; MN, Minnesota; IN, Indiana; KS, Kansas; KY, Kentucky; VA, Virginia; NY, New York; AL, Alabama; MD, Maryland; AR, Arkansas; GA, Georgia; WI, Wisconsin.

**Table 5.** Land improvement and vegetated buffer impacts on reduction in phosphorus loss.

| Purpose | State | Soil Series | Crop | P application (kg ha$^{-1}$) | Tillage | STP (mg kg$^{-1}$) | Study Type | Highlights | Reference |
|---|---|---|---|---|---|---|---|---|---|
| **Conservation practices impact on managing P loss in runoff and its relation to sediment P and DP.** | IA | Monona, Ida, and Napier SiL | C, P | 39 and 97 | na | na | Terraces | Average loss of DP was 0.049 and inorganic P (IP) was 0.085 kg ha$^{-1}$ yr$^{-1}$ from terraced fields whereas 0.171 (DP) and 1.05 (IP) kg ha$^{-1}$ yr$^{-1}$ for fields without terraces | Schuman et al. [251] |
| **Influence of levelled terraces and contour planted corn on water quality** | IA | Marshall SiL, Judson SiL, Monnona SiL | C | na | na | na | Terraces | na | Burwell et al. [252] |
| **Nutrient losses in water from terraced continuous row cropping system** | IA | Fayette silt, Clarion loam, Sharpsburg SiC, Floyd loam | C | 17 to 43 | MP | 7–42 | Terraces | Generally total P losses in runoff were 0.44 to 1.06 kg ha$^{-1}$ and were correlated to sediment loss | Hanway and Laflen [253] |
| **Watershed budgets of N and P were calculated using crop removal, surface runoff loss, deep percolation and subsurface discharge** | IA | Marshall SiL, Judson SiL, Monnona SiL | C | 17 to 48 | CT | na | Terraces | Terraced watersheds had 0.11 to 0.46 kg ha$^{-1}$ yr$^{-1}$ less total P loss compared to contour watersheds | Burwell et al. [254] |
| **Nitrogen and P losses for three seasonal runoff and erosion periods** | IA | Marshall SiL, Judson SiL | C | 39 and 97 | MP, DT | na | Terraces | Terraced watersheds had 0.019 to 0.048 kg ha$^{-1}$ yr$^{-1}$ DP loss whereas contour watersheds had 0.022 to 0.045 kg ha$^{-1}$ yr$^{-1}$ | Alberts et al. [255] |
| **N and P loss in surface and subsurface water for a 10-yr period from terrace and contour till fields** | IA | Marshall SiL, Judson SiL | C | 36 and 97 | MP, DT | na | Terraces | P loss in surface runoff was <2% of applied fertilizer and was highest for tilled fields | Alberts and Spomer [256] |

Table 5. *Cont.*

| Purpose | State | Soil Series | Crop | P application (kg ha$^{-1}$) | Tillage | STP (mg kg$^{-1}$) | Study Type | Highlights | Reference |
|---------|-------|-------------|------|------------------------------|---------|---------------------|------------|------------|-----------|
| design and potential of blind inlets to improve water quality compared to tile risers | IN | na | C, S, O, W | na | na | na | Blind inlets | Reduction of 65–72% in DP and 50–78% in total P loading was reported by replacing tile risers with blind inputs | Smith and Livingston [257] |
| Blind inlets and tile riser were evaluated for suspended sediment and P loads from drainage water | IN, MN | na | A, C, S, O, W | 20–54 | na | na | Blind inlets | Total P and DP loads were 66% and 50% less for the blind inlets compared to tile risers | Feyereisen et al. [99] |
| effects of inclusion of a cover crop on nutrient loss | KS | Smolan SiCL | C, CC | 36 | NT | na | Cover crops, terraces | Out of 2 yr runoff DP loss was reduced only in 1 yr in the terraces with cover crops | Abel [76] |
| Effects of inclusion of a CC in a corn-soybean rotation on nutrient loss, soil health, and crop yields in a terraced field | MO | Putman SiL | C, CC | 28 | NT | 80–90 | Cover crops, terraces | Terraces with cover crops did not decreased cumulative total P loss | Adler et al. [98] |
| Nutrient (C, N, and P) concentration changes in surface runoff and shallow groundwater | MD | SL | na | na | na | na | Riparian buffers | na | Peterjohn and Correll [258] |
| Performance of vegetative filter strips (VFS) of different lengths for the removal of sediment, nitrogen (N), and phosphorus (P) from cropland runoff | VA | Groseclose SiL | Orchard grass | 49 | CT | na | In-Field vegetative buffers | Reduction by 70–84% of the incoming suspended solids, 61–79% of the incoming P, and 54–73% of the incoming N | Dillaha et al. [259] |

**Table 5.** *Cont.*

| Purpose | State | Soil Series | Crop | P application (kg ha$^{-1}$) | Tillage | STP (mg kg$^{-1}$) | Study Type | Highlights | Reference |
|---|---|---|---|---|---|---|---|---|---|
| **Performance of vegetated filter strips of different lengths in nutrients and sediments reduction from agricultural runoff.** | MD | Woodstown SL | Ky-31 fescue | 114 | na | na | In-Field vegetative buffers | 66%, 0% and 27% reduction in TSS, TN, and TP from runoff masses | Magette et al. [260] |
| **Natural and planted VFS effectiveness in reducing sediment and nutrient losses** | NC | Cecil, Georgeville | Fescue, shrubs, trees | na | na | na | In-Field vegetative buffers | 50–80% reduction in runoff load, 80% reduction in sediment loss, 50% and 80% reduction in TP and Phosphate-P | Daniels and Gilliam [261] |
| **Performance of poultry treated VFS of varying filter strip length in reducing nutrient losses from varying pollutant source runoff** | AR | Captina SiL | Manured treated fescue, P | 60 | na | 60 | In-Field vegetative buffers | 22–82% phosphate-P and 21–66% TP reduction from runoff by VFS | Srivastava et al. [262] |
| **Effectiveness of natural riparian grass buffer strips in removing sediment, atrazine, nitrogen and phosphorus from surface runoff** | KY | Maury SiL | na | na | na | na | In-Field vegetative buffers | na | Barfield et al. [263] |
| **Comparison of switch grass and cool-season grass strips of different width in sediment and nutrients reduction** | IA | Coland | Switchgrass, bromegrass, timothy, fescue | na | ns | na | In-Field vegetative buffers | Filter strips removed 66–77% sediment, 37–52% TP and 34–43% phosphate P, depending upon filter strip width. Switchgrass filter strip removed more TP and phosphate than cool season grass filter strips | Lee et al. [264] |

**Table 5.** *Cont.*

| Purpose | State | Soil Series | Crop | P application (kg ha$^{-1}$) | Tillage | STP (mg kg$^{-1}$) | Study Type | Highlights | Reference |
|---------|-------|-------------|------|------------------------------|---------|---------------------|------------|------------|-----------|
| **Effects of using different vegetation types (mixed grasses, trees, shrubs) and width in buffer filter strip on runoff reduction** | NE | Sharpsburg SiCL | M, S, switchgrass, tall fescue, bush honeysuckle, golden currant eastern cottonwood, silver maple | na | CT | na | In-Field vegetative buffers | 76–93% and 55–79% reduction in runoff and total P by buffer strips | Schmitt et al. [265] |
| **Reduction in P and N transport by narrow switchgrass hedges following application of manure and inorganic fertilizer under NT and disked till.** | IA | Monona SiL | C, switchgrass | 25.8 | NT, DT | 27–101 | In-Field vegetative buffers | In NT, buffer reduced runoff DP, BAP, PP, TP by 47%, 48%, 38%, and 40% than control. In disked, buffer reduced runoff DP, BAP, PP, TP by 21%, 29%, 43%, and 38% than control. | Eghball et al. [266] |
| **Effectiveness of multiple species riparian buffers in reducing N and P runoff losses** | IA | Coland SiCL, Clarion loam | C, S, switchgrass, woody plant buffer | na | na | na | Riparian buffers | Switchgrass buffer and switchgrass-woody buffer reduced sediment loss by 70% and >92%, respectively. Switchgrass buffer removed TP and phosphate-P by 72% and 44% during 2-hr rainfall simulation at 25 mm/hr and by 46% and 28% during 1-hr rainfall simulation at 69 mm/h. Switchgrass-woody buffer removed TP and phosphate-P by 93% and 85% during 2-hr rainfall simulation at 25 mm/hr and by 81% and 35% during 1-hr rainfall simulation at 69 mm h$^{-1}$ | Lee et al. [267] |

**Table 5.** *Cont.*

| Purpose | State | Soil Series | Crop | P application (kg ha$^{-1}$) | Tillage | STP (mg kg$^{-1}$) | Study Type | Highlights | Reference |
|---|---|---|---|---|---|---|---|---|---|
| **Effectiveness of agroforestry, and contour legume-grass filter strips in reducing sediment and nutrient loss from watershed planted with corn-soybean crops.** | MO | Putnam SiL, Kilwinning SiL, Armstrong loam | C, S, redtop, brome grass, pin oak, swamp white oak, bur oak | 18–22 | NT | na | In-Field vegetative buffers | Contour strip reduced runoff, erosion, TP by 10%, 19%, and 8%, respectively. Agroforestry reduced runoff and TP by 1% and 17%. | Udawatta [268] |
| **Effectiveness of multispecies buffer in reducing sediment, nitrogen, and phosphorus from runoff** | IA | Clarion, Coland | Switchgrass, woody buffer, chokecherry cherry, wild plum, red osier dogwood, ninebark | na | na | na | Riparian buffers | Switchgrass buffer removed sediment, total-N, nitrate-N, TP, and phosphate-phosphorus from runoff by 95%, 80%, 62%, 78%, and 58%, respectively. switchgrass/woody buffer removed sediment, total-N, nitrate-N, TP, and phosphate-phosphorus by 97%, 94%, 85%, 91%, and 80%, respectively. | Lee et al. [269] |
| **Variability of N, P, and chloride movement and loads in surface runoff in a grass filter strip, a mature riparian forest, and a managed riparian buffer** | GA | Alapaha LS, Tifton LS | C, Peanut, millet, bermudagrass, Bahia grass, perennial ryegrass, slash pine, long leaf pine, yellow poplar, swamp black gum | na | CT | na | Riparian buffers | 27% reduction in TKN, 63% reduction in sediment P; on average ~65% reduction in all nutrient load | Lowrance and Sheridan [270] |
| **Effectiveness of VBSs on surface runoff water quality** | IL | Hosmer SiL | C, S, giant cane, Kentucky bluegrass, orchard grass, bareground | 37 | CT | na | In-Field vegetative buffers | All VBSs reduced total P in runoff by 0.84–1.16 mg L$^{-1}$ than corn | Singh et al. [108] |

Abbreviations: BAP, bioavailable phosphorus; CT, conventional tillage; DP, dissolved phosphorus; DT, disk tillage; MP, moldboard plow; na, not available; NT, no-tillage; P, phosphorus; PP, particulate phosphorus; TKN, total Kjeldahl nitrogen; TN, total nitrogen; TP, total phosphorus; TSS, total suspended solids; VBS, vegetative buffer strips; VFS, vegetative filter strips.

**Table 6.** Conservation practices for managing tile drain water.

| Purpose | State | Soil Series | Crop | P application | Tillage | STP (mg kg$^{-1}$ or kg ha$^{-1}$*) | Study type | Highlights | Reference |
|---|---|---|---|---|---|---|---|---|---|
| **Impact of cropping systems with tile drains on nitrate and phosphate content of water** | VT | Cabot SiL | P, A, C | na | na | na | Subsurface tile water monitoring | Vertical and lateral movement of P through the soil to subsurface drains was not reported | Benoit [271] |
| **Monitoring of nutrient losses in subsurface drainage water** | IA | Clarion loam, Webster SiCL | O, C, S | na | na | na | Subsurface tile water monitoring | Annual loss of DP ranged between 0 to 0.04 kg ha$^{-1}$ | Baker et al. [272] |
| **Monitoring of nutrient and sediment losses for 10 yr in deep and shallow tile drainage** | OH | SiC | A, O, C, S | na | CT | na | Subsurface tile water monitoring | Average P loss was between 0.8 to 1.2 kg ha$^{-1}$ | Schwab et al. [273] |
| **Land clearing and improved drainage effects on the drainage water quantity and quality** | NC | Portsmouth fSL, Wasda muck | C, S | na | na | na | Controlled drainage structures | Annual total P loss in tiles ranged between 0.2 to 7.6 kg ha$^{-1}$ | Gilliam and Skaggs [274] |
| **Monitoring of pesticide, nutrient, and sediment concentrations in subsurface tile drains for 3 yr** | IN | Clermont SiL | C | na | CP | na | Subsurface tile water monitoring | Annual loss of DP averaged 0.04 kg ha$^{-1}$ | Kladivko et al. [275] |
| **Evaluation of nutrient loss between conventional and controlled drainage** | NC | na | na | na | na | na | Controlled drainage structures | P reduction of 50% by using controlled drainage was reported | Evans et al. [276] |
| **P export patterns from tile drains and estimated P output from fields and watersheds** | IL | Drummer SiCL | C, S | 42 kg P ha$^{-1}$ yr$^{-1}$ | na | na | Subsurface tile water monitoring | Dissolved P export varied between 0.18 to 0.79 kg P ha$^{-1}$ yr$^{-1}$ | Xue et al. [277] |

**Table 6.** *Cont.*

| Purpose | State | Soil Series | Crop | P application | Tillage | STP (mg kg$^{-1}$ or kg ha$^{-1}$*) | Study type | Highlights | Reference |
|---------|-------|-------------|------|---------------|---------|-------------------|------------|------------|-----------|
| **Measuring loss of N, P, and fecal indicator bacteria in tile drainage and change in STP and STK in fields applied with dairy manure and urea applied fields** | MN | Webster CL | C | 36 to 68 kg ha$^{-1}$ | CT | 30 | Subsurface tile water monitoring | Total P and DP losses were very small and averaged 31 and 10 g ha$^{-1}$ yr$^{-1}$ | Randall et al. [278] |
| **Tillage and fertilizer source interactions on sediment, N, and P loss from surface and subsurface tile drains** | MN | Webster CL | C | 53, 69, and 86 kg ha$^{-1}$ | MP, ReT | 12 to 26 | Subsurface tile water monitoring | Dissolved P loss in ridge till varied between 12 to 140 g ha$^{-1}$ whereas for moldboard varied between 0.1 to 4.6 g ha$^{-1}$ | Zhao et al. [245] |
| **Dominant P forms and primary P transport pathways for tile drained watersheds** | IL | na | C, S | ~50 kg ha$^{-1}$ | na | na | Subsurface tile water monitoring | Dissolved P concentrations were highest (1.25 mg L$^{-1}$) when precipitation event followed widespread application of P fertilizer on frozen soils. | Gentry et al. [117] |
| **Evaluation of agricultural management practices for DP losses in subsurface tile flow and surface runoff** | IL | Drummer, Flanagan SiCL, Sabina and Xenia SiL | C, S | 44 to 74 kg ha$^{-1}$ | ReT, NT | na | Subsurface tile water monitoring | Average flow-weighted soluble P concentrations in subsurface flow ranged between 0.09 to 0.19 mg L$^{-1}$ | Algoazany et al. [60] |

**Table 6.** *Cont.*

| Purpose | State | Soil Series | Crop | P application | Tillage | STP (mg kg$^{-1}$ or kg ha$^{-1}$*) | Study type | Highlights | Reference |
|---|---|---|---|---|---|---|---|---|---|
| **Evaluated of yield, drain flow, and NP loads through subsurface drainage from free-drainage and controlled drainage** | MN | Millington SiL | C, S | 67 and 138 kg ha$^{-1}$ | na | 17 to 21 [1] | Controlled drainage structures | Total P and DP losses were reduced by 50% and 63% with controlled drainage compared to free drainage | Feset et al. [120] |
| **Design of P removal structure and its P removal efficiency by monitoring inflow and outflow water.** | OK | na | na | na | na | na | Removal structure for P | 54% of P was removed from inflow water and life of stricture was estimated ~17 m | Penn et al. [279] |
| **Calculation of P loads from discharge data for 8 yr** | OH | Bennington SiL, Pewamo CL | C, S | na | CP, NT | na | Subsurface tile water monitoring | Tile drainage accounted for 47% of the discharge, 48% of the dissolved P, and 40% of the total P exported from the watershed. | King et al. [280] |
| **Phosphorus loading via subsurface tile** | IN | na | na | na | na | na | Subsurface tile water monitoring | 49% of DP and 48% of total P losses were reported to occur via tile discharge | Smith et al. [281] |
| **Differences in DP losses from controlled drainage and free drainage** | MO | Putnam SiL | C | 35 and 78 kg ha$^{-1}$ | VT, TO | 71 to 88* | Controlled drainage structures | Dissolved P loss in tile drain water was reduced by 80% with controlled drainage compared to free drainage | Nash et al. [121] |
| **P loss in tile drainage water, movement and accumulation of P in topsoil after long-term application of poultry manure** | IA | Nicollet; fine loamy, Canisteo, Harps | C, S | na | na | 20 to 80 | Subsurface tile water monitoring | Average DP concentration in tile drainage ranged between 0.01 and 0.02 mg L$^{-1}$ | Hoover et al. [282] |

**Table 6.** *Cont.*

| Purpose | State | Soil Series | Crop | P application | Tillage | STP (mg kg$^{-1}$ or kg ha$^{-1}$*) | Study type | Highlights | Reference |
|---|---|---|---|---|---|---|---|---|---|
| **Subsurface tile P drainage loss in different cropping systems** | IA | Webster, Nicollet | C, S, CC | 61 to 94 kg P ha$^{-1}$ | DT | na | Subsurface tile water monitoring with cover crops | Dissolved P ranged between 0.02 and 0.04 kg ha$^{-1}$ and were not significantly affected by cropping systems | Daigh et al. [283] |
| **Measured Annual Nutrient loads from Agricultural Environments (Manage) database was evaluated for water quality associated with P management strategies** | IL | na | na | na | na | na | Subsurface tile P loss review | Generally, less than 2% of applied P was lost in drainage water and no-till significantly increased drainage DP loads compared with conventional tillage | Christianson et al. [118] |
| **Effectiveness of winter cover crops in reducing N and total P loading from tile drained agricultural watershed** | IL | na | C, S, CR, R | na | na | na | Subsurface tile water monitoring with cover crops | CC significantly reduced n of total P loading in baseflow were reported in watershed planted with cover crops | Bruening [77] |
| **Effectiveness of woodchip bioreactors and a P adsorption structure in removing N and DP from subsurface drainage water** | SD | na | C, S, W | na | na | na | Bioreactor and P adsorption structure | Dissolved P reduction: 10% to 9% P removal rates: 2.2 to 183.7 g m$^{-3}$ d$^{-1}$ | Thapa [135] |

Abbreviations: CP, chisel plow; CT, conventional tillage; DP, dissolved P; DT, disk tillage; MP, moldboard plow; na, not available; NT, no-tillage; P, phosphorus; ReT, reduced tillage; TO, tilloll; VT, vertical tillage.

**Table 7.** Phosphorus removal by wetlands and drainage ditches.

| Purpose | State | Soil Series | Hydraulic Load (mm d$^{-1}$) | Detention Time | Study Type | Highlights | Reference |
|---|---|---|---|---|---|---|---|
| Annual patterns in hydrology, P circulation, and sediment dynamics | IL | na | na | | Wetlands | Reduction in P was 10 times at outflow compared to in flow | Mitsch et al. [284] |
| Evaluation of a constructed wetland for controlling non-point source pollution | IL | na | 1.4–86 | na | Wetlands | P removal efficiencies ranged from 60% to 100% in summer and from 27% to 100% in winter | Kadlec and Hey [145] |
| Potential of a constructed wetlands receiving tile drain water for removing N and P | IL | Colo SiCL | 17–30 | 22.38 | Wetlands | Total P removed varied from −76 to 8.5 kg ha$^{-1}$ yr$^{-1}$ | Kovacic et al. [149] |
| Effectiveness of wetland to reduce N and P from agricultural drainage water | IL | na | na | na | Wetlands | Reduction in DP concentration and load was not significant | Miller et al. [150] |
| Monitoring of water, nutrients, and sediment flux into and out of the wetland | MD | Othello, Mattapex | 12–20 | 12–19 | Wetlands | Total P removed varied between −2.8 to 18 kg ha$^{-1}$ yr$^{-1}$ | Jordan et al. [151] |
| Potential of restored forested riparian wetland buffer system for removal of N and P from water | GA | Alapaha LS, Tifton LS | na | na | Wetlands | Retention rates of DP and total P by wetland were 66%. | Vellidis et al. [146] |
| A prairie pothole restored wetland was monitored for P removal | MN | na | na | na | Wetlands | Total P removed varied from 1 to 3 kg ha$^{-1}$ yr$^{-1}$ | Magner et al. [285] |

**Table 7.** *Cont.*

| Purpose | State | Soil Series | Hydraulic Load (mm d$^{-1}$) | Detention Time | Study Type | Highlights | Reference |
|---|---|---|---|---|---|---|---|
| **Comparison of annual loading and removal of P by a wetland receiving urban water and a wetland receiving rural water** | WA | na | 620–720 | 3.3–20 | Wetlands | Total P removed varied from 4.4 to 30 kg ha$^{-1}$ yr$^{-1}$ | Reinelt and Horner [286] |
| **Mitigation capacity of agricultural drainage ditches by measuring DP and total P** | MS | Chenneby SiL | na | na | Drainage diches | Average P removed was 1.43 kg ha$^{-1}$ yr$^{-1}$ | Kröger et al. [287] |
| **Comparison of a vegetated versus non-vegetated agricultural drainage ditch in reducing nutrient concentrations and loads** | MS | Sharkey, Dundee | na | na | Drainage diches | No reduction in DP however 36–71% reduction in inorganic P was reported | Moore et al. [288] |

Abbreviations: DP, dissolved P; na, not available; P, phosphorus.

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
