# Peer review of "Managing Phosphorus Loss from Agroecosystems of the Midwestern United States: A Review"

_agronomy, doi:10.3390/agronomy10040561_

Round 1

Reviewer 1 Report

Comments to the manuscript: Singh et al. Managing phosphorus loss from agroecosystems: A review

General comments

This is a Nice over view and an interesting review describing a comprehensive amount of studies and pulling together the results. However, it is a rather local approach and it does not include all international studies relevant for each topic. It would be of higher interest if it described the situation in a specific climate region instead of referring to only the Mid-western USA. The authors claim that studies are missing, but they do not refer to all the studies that have been carried out outside US under similar conditions as in the Midwest. The soil P desorption and adsorption and the amendments could very well be referred to other parts of the world. Most of the P source processes in soil could as well refer to literature outside this narrow area in the Midwest.

I agree with the focus on phosphorus status of soils instead of only focusing on erosion as a measure to reduce P loss. However, it may be worth also to include other measures on erosion in addition to tillage, e.g. grassed water ways and grasstrips dividing the slope. These are important in some regions. Please evaluate whether they may be important in the Mid-west.

Specific comments.

Key words: WASCoBs is a strange keyword. I don’tknow hwat it means.

Line 17: Please write “Our review shows that cover crops, reduced tillage….aso.” to make it clear where your results start.

Line 32: I’m not sure that P is the most limiting nutrient; I think that N is even more limiting to growth. I suggest the first sentence focus on the limited amounts of P in the world. Write limited instead of limiting.

Line 38: Be careful with wording; I do not think that P levels are susceptible to P loss; high soil P levels have a risk of P loss.

Line 44: Poor experimental design is not a reason for poor performance of BMPs. It may rather be a reason for not being able to prove the efficiency.

Line 49: If this review is only of interest for Midwestern United States it may not suit an international journal. Is it of value for other areas? Could you mention the climatic zone instead of the states?

Line 57: What is IL? Other names are also mentioned which are too local, e.g. Sanborn.

Line 64: This must be a local case, please specify which country you are referring to.

Line 72: There are several STP-methods. Please specify which method you are using. The same throughout the manuscript

Line 79: What kind of soil conditions were improved by strip tillage? Was it just the uptake of P that was improved, then write “…resulted in better P uptake…”.

Line 81: Could you add a sentence about the yield response at high STP.

Line 94-96: Reformulate this sentence

Line97: Not easy to follow. Is there first a buildup and then a decline in STP?

Line 109-110: reformulate this sentence

Line 116: Please refer to some of the studies outside the MidWestern US.

Line135: What are P enhancers, Avail and P2O5-Max? Please describe.

Line 152: You write that “Some of the common BMPs…”, I suggest you delete “Some of the common”

Line 158: Delete “Most of the” since there are many studies in other parts of the world.

Line 161: Did Algoazany et al. study total P or DRP? Please specify

Line 162: Specify if Yuan et al. studied No-till systems

Line 172-175: This sentence is difficult to understand. Please rewrite.

Line 175-176: I don’t think it was actually the fertilizer which was lost, but amounts corresponding to the fertilizer amount.

Line176: The last sentence can be moved to the method section. If the whole study is based on MANAGE you may write a method section describing this database.

Line 181: Specify whether Villamil et al. harvested the cover crop

Line 227: Describe the measure that was implemented in reference 89

Line 248: To me it seems illogic that the efficiency is improved if they are not protected. Should “not” be deleted?

Line 261: It is unclear to me what a blind inlet is. Please explain.

Line 278: start line with ”reduces”

Line 278: Place a “.” after [104, 105]

Line 296: How did Stutter et al. explain their results? Same as Roberts et al.?

Line 297: What is P solubility indices? Please explain

Line 308: delete Roberts et al. in the beginning.

Line 315: harvesting of the buffer could be including as one of the most common management methods

Line 272-320: in this section I’m missing a discussion of the importance of the topography and slope of the buffer itself. They are not always flat

Line 327: Please be more specific on how many results the watershed has provided; more specific than just “some”

Line 332: Suggest to write “P loading to subsurface tiles is impacted by preferential flow in soil, soil P …aso.

Line 346: Please give a reference after “…field.”

Line 372: Please describe first what a bioreactor is. This is not a concept used in my country. Treatment of N and P either in tile drainage water, surface water or small streams requires a high hydraulic capacity of the filter material. Please put more focus on the hydraulic capacity in this section.

Line 523: I think in many areas crop production can be sustained even with lower P application and I don’t think it ought to be a challenge, but required common understanding. Suggest to remove the sentence “Sustaining the present…..challenge”.

Line 528-529: If knowledge is limited in the Mid-west, would it be possible to use knowledge from other regions on this topic?

Figure 1: Please divide this into separate Figures. It is difficult to read when everything is in one Figure.

Author Response

File is attached for answers to the reviewer's comments.

Reviewer 2 Report

To the authors:

I much enjoyed reading this review of managing P loss from agroecosystems. The authors have brought together an impressive number of studies albeit from a limited but important agricultural area. The text is well written, well referenced and easy to follow. They identify specific issues related to P retention and loss and where appropriate describe research topics which need to be followed.

It is my experience, albeit in more general aquatic science, that P is often less studied than N and thus this review seems to me well worthwhile.

I have identified some very trivial minor changes and would otherwise recommend publication essentially as is.

Line 36 have not has

Line 220 Could the authors define what are ‘water and sediment control basins’ or is that obvious to the target readers?

Line 314 remove needed

Author Response

We would like to thank the reviewers for their review and comments and helping us improve this manuscript for publication. Responses are provided in the red text following each comment and changes are made using track changes in the accompanying document. 

Reviewer Comments:

I much enjoyed reading this review of managing P loss from agroecosystems. The authors have brought together an impressive number of studies albeit from a limited but important agricultural area. The text is well written, well referenced and easy to follow. They identify specific issues related to P retention and loss and where appropriate describe research topics which need to be followed.

It is my experience, albeit in more general aquatic science, that P is often less studied than N and thus this review seems to me well worthwhile.

 I have identified some very trivial minor changes and would otherwise recommend publication essentially as is.

 Line 36 have not has

 The wording is corrected.

Line 220 Could the authors define what are ‘water and sediment control basins’ or is that obvious to the target readers?

We added information about the “water and sediment control basins”.

Water and sediment control basins are designed along minor slopes or watercourses. They detain precipitation runoff during storm events and slowly discharge this water through a stable outlet, which reduces streamflow by 5%. Sediment in the runoff is deposited in the basin, removing 80% of total suspended solids in runoff, reducing the phosphorus load by 85% and improving downstream water quality. These basins eliminate gully erosion and can also remove pollutants such as nitrogen, metals, bacteria and hydrocarbons. The dry basin between storm events maintains land function and farmability and vegetation along the basin edge provides wildlife habitat.

Line 314 remove needed

“needed” is deleted.